# HUMOF: HUMAN MOTION FORECASTING IN INTERACTIVE SCENES

**Caiyi Sun**$^{1,2*\dagger}$ **Yujing Sun**$^{2,3*}$ **Xiao Han**$^1$, **Zemin Yang**$^1$,
**Jiawei Liu**$^4$, **Xinge Zhu**$^5$, **Siu-Ming Yiu**$^{2\ddagger}$ **Yuexin Ma**$^{1\ddagger}$
$^1$ShanghaiTech University
$^2$The University of Hong Kong
$^3$Digital Trust Centre, Nanyang Technological University
$^4$Sun Yat-sen University
$^5$The Chinese University of Hong Kong
`caiyisun.scy@gmail.com, mayuexin@shanghaitech.edu.cn`

## ABSTRACT

Complex dynamic scenes present significant challenges for predicting human behavior due to the abundance of interaction information, such as human-human and human-environment interactions. These factors complicate the analysis and understanding of human behavior, thereby increasing the uncertainty in forecasting human motions. Existing motion prediction methods thus struggle in these complex scenarios. In this paper, we propose an effective method for human motion forecasting in dynamic scenes. To achieve a comprehensive representation of interactions, we design a hierarchical interaction feature representation so that high-level features capture the overall context of the interactions, while low-level features focus on fine-grained details. Besides, we propose a coarse-to-fine interaction reasoning module that leverages both spatial and frequency perspectives to efficiently utilize hierarchical features, thereby enhancing the accuracy of motion predictions. Our method achieves state-of-the-art performance across four public datasets. The source code will be available at `https://github.com/scy639/HUMOF`.

## 1 INTRODUCTION

Human motion forecasting is essential across a wide range of applications, including surveillance, healthcare, autonomous driving, and human-robot interaction. The ability to accurately anticipate human behavior in dynamic environments is key to enhancing system safety, operational efficiency, and user experience. However, this task presents significant challenges, including the inherent complexity and variability of human motion, as well as the impact of diverse environmental factors.

In early times, many works predominantly addressed the task of human motion prediction by using simple representations of environmental states. For example, some methods (Zhang et al., 2023; Xu et al., 2023b; Ma et al., 2022; Xu et al., 2023a; Gao et al., 2023; Aksan et al., 2021; Wang et al., 2024; Su et al., 2021; Tang et al., 2023) rely solely on past human actions to predict their future motions, while others (Cao et al., 2020; Mao et al., 2022; Scofano et al., 2023; Zheng et al., 2022; Xing et al., 2025) integrate static scene features into the network all at once. However, these approaches struggle to adapt to real-world applications, where dynamic environmental constraints play a crucial role. Actually, to better predict how humans respond to dynamic environments, it is essential to consider the interaction influence. Some works (Wang et al., 2021; Guo et al., 2022b; Vendrow et al., 2022; Saadatnejad et al., 2024; Gao et al., 2024a;b; Xu et al., 2023c; Peng et al., 2023; Xiao et al., 2025) have started addressing motion prediction in challenging multi-person scenarios, using attention mechanisms to implicitly model the human-human interaction. However, these works overlook the dynamic relationship between humans and the nonhuman environment, which is equally critical for accurate motion forecasting in real scenes.

---

$^*$Equal contribution.
$^\dagger$Work done during internship at ShanghaiTech University.
$^\ddagger$Corresponding authors.

In fact, real-world environments are inherently complex and dynamic, where existing frequent human-human interactions, e.g., engaging in conversation, approaching others, or avoiding collisions, as well as human-scene interactions, e.g., sitting on stairs, lying on a bed, as shown in Figure 1. It is important to model all human-related interactions in one framework for more accurate human motion forecasting. Although (Mueller et al., 2024) made the first attempt to address the problem under this setting, it decouples feature extraction for interacting humans and scenes, fails to fully capture the interaction features, and relies on predefined semantic labels for the scene. As a result, its prediction performance is limited, and it is not practical for real-world applications. The main challenge for forecasting human motion in a realistic and dynamic environment is twofold. Given the vast array of diverse, multi-level interactions between humans and their surroundings, as well as between individuals, *how can we design effective representations to capture these complex interactions?* Moreover, even with well-encoded interaction representations, *how can we leverage them effectively to enhance prediction accuracy?*

In this paper, we have addressed the above two challenges and propose a novel method, named **HUMOF**, for HUman MOtion Forecasting in complex dynamic scenes. It effectively models human kinematics and dynamics, spatial environment states, temporal information, and the most crucial interaction features, offering significant potential as a world model for human motion. In particular, we introduce a **Hierarchical Interaction Representation** to effectively capture complex and valuable interaction features. The hierarchical representation manifests in several dimensions: (1) It includes both human-human interaction modeling and human-scene interaction modeling; (2) It captures interactions through explicit representations, i.e., interactive distances, and implicitly learns interaction features through the network; (3) It integrates both high-level semantic interaction features and low-level geometric interaction features. Furthermore, to fully utilize the hierarchical representation for enhancing human motion prediction, we design a **Coarse-to-Fine Interaction Reasoning Module**. Specifically, to encourage the model to focus on global environmental understanding while minimizing interference from noisy low-level environmental information and high-frequency details in the earlier stages, and refine fine-grained details in the later stages, we implement the coarse-to-fine mechanism from two perspectives: (1) In spatial perspective, through our coarse-to-fine injection strategy, high-level features are injected into early Transformer layers for semantic understanding of human actions, while low-level features are introduced in later Transformer layers to perceive geometric details; (2) In frequency perspective, our DCT rescaling mechanism suppresses the updating of high-frequency components of human motion in earlier layers, and progressively encourages the model to focus on low-frequency details in later stages. Extensive experimental results demonstrate that our method achieves state-of-the-art performance on four public datasets, and ablation studies show the effectiveness of our detailed designs. Our contributions are summarized as follows:

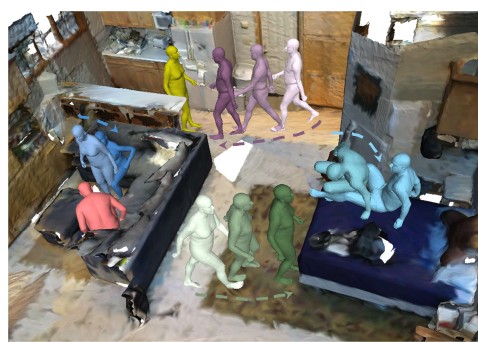

Figure 1: Real dynamic scenes involve complex human-human and human-scene interactions. We propose to predict human motions under such challenging settings, where existing methods struggled.

- We present an effective method for human motion prediction in dynamic environments, involving both human-human and human-scene interactions, achieving state-of-the-art performance in various dynamic scenarios.

- We introduce hierarchical interaction feature representation to achieve a comprehensive understanding of human-human and human-scene interactions.

- We propose a coarse-to-fine interaction reasoning module to fully leverage hierarchical interactive features to enhance prediction accuracy.

## 2 RELATED WORK

**Single-Person Human Motion Prediction**    Early works mainly consider the own kinematic and dynamic influence on future human motions and predict the motion for a single person (Zhang et al.,

2023; Xu et al., 2023b; Ma et al., 2022; Xu et al., 2023a; Gao et al., 2023; Aksan et al., 2021; Wang et al., 2024; Su et al., 2021; Tang et al., 2023). Many approaches (Fragkiadaki et al., 2015; Jain et al., 2016; Martinez et al., 2017; Liu et al., 2022) relied on Recurrent Neural Networks (RNNs) to capture temporal dependencies, overlooking spatial relationships. More recent methods have shifted towards Graph Convolutional Networks Li et al. (2022); Chen et al. (2020); Dang et al. (2021), Temporal Convolutional Networks (Sofianos et al., 2021), and Transformers Mao et al. (2020); Cai et al. (2020); Aksan et al. (2021); Xu et al. (2023a), aiming to capture complex spatial-temporal relationships. However, these methods primarily concern personal situations to predict future motions, limiting the application in real-world scenarios.

**Scene-Aware Human Motion Prediction** Recent advancements (Cao et al., 2020; Mao et al., 2022; Scofano et al., 2023; Zheng et al., 2022; Xing et al., 2025) have started incorporating scene context into human motion prediction tasks. Some approaches (Cao et al., 2020) represented scenes as 2D images, but struggled when handling occlusions and failed to maintain consistency between local and global motion. GIMO (Zheng et al., 2022) attempted to enhance prediction accuracy by incorporating eye gaze; ContactAware (Mao et al., 2022) leveraged a contact map to encode human-scene relationships; STAG (Scofano et al., 2023) proposed a three-stage approach that sequentially processes contact points, trajectories, and poses. MutualDistance (Xing et al., 2025) offered an explicit human-scene interaction model using mutual distance. Although these methods have effectively modeled human-scene interactions, they focus on static scenes, neglecting dynamic social interactions between humans.

**Social-Aware Human Motion Prediction** Recent studies (Adeli et al., 2020; 2021; Wang et al., 2021; Guo et al., 2022b; Vendrow et al., 2022; Tanke et al., 2023b; Saadatnejad et al., 2024; Gao et al., 2024a;b; Xu et al., 2023c; Peng et al., 2023; Jeong et al., 2024; Xiao et al., 2025) in multi-person human pose forecasting focus mainly on modeling human interactions in group scenarios. Most recently, Transformers (Wang et al., 2021; Guo et al., 2022b; Vendrow et al., 2022; Saadatnejad et al., 2024; Gao et al., 2024a;b; Xu et al., 2023c; Peng et al., 2023; Xiao et al., 2025) are popular for this task due to their strong learning capabilities: T2P (Jeong et al., 2024) sequentially predicts global trajectory and local pose; IAFormer (Xiao et al., 2025) proposed to learn amplitude-based interactions and prior knowledge. However, methods in this category overlook the importance of scene information. A recent work (Mueller et al., 2024) uses the diffusion model for long-term motion generation considering both static scene and motion of other individuals. However, it only implicitly encodes the scene and other individuals, without explicit modeling of human-to-scene and human-to-human interaction. Additionally, it treats the scene as a set of discrete objects with semantic tags, relying on ground-truth segmentation results, which limits its applicability in real-world scenarios involving raw sensor data.

## 3 METHODOLOGY

The key challenge in forecasting human motion within complex dynamic environments lies in effectively encoding and leveraging the involved human-human and human-scene interactions. Hence, we, on one hand, propose a hierarchical approach to comprehensively encode these representations (Figure 3ab), and on the other hand, present a Coarse-to-Fine Interaction Reasoning Module (Figure 3c) to fully leverage the representations.

**Problem Definition.** The task is to predict a person's future motion given their past motion, the point cloud of static scene elements, and the past motion of other individuals in the vicinity.
**The input of our model** includes three parts: **1).** A historical motion sequence of the target person $\mathbf{X}^{1:H}$ where $\mathbf{x}_j = \{\mathbf{x}_j^1, \cdots, \mathbf{x}_j^H\} \in \mathbb{R}^{H \times 3}$ represents the motion of $j_{th}$ joint, with each $\mathbf{x}_j^t$ corresponding to the 3D coordinates of a joint at $t_{th}$ frame; **2).** The scene's 3D point cloud $\mathcal{S} = \{s_1, \cdots, s_N\}$ with $N$ points; And **3).** the historical motion sequence $\mathcal{Y}^{(\mathbf{k})} = [\mathbf{y}_1^{(k)}, \cdots, \mathbf{y}_J^{(k)}] \in \mathbb{R}^{J \times H \times 3}$ of the $k_{th}$ $(k \in [1, K])$ interactive person in the scene, which also consists

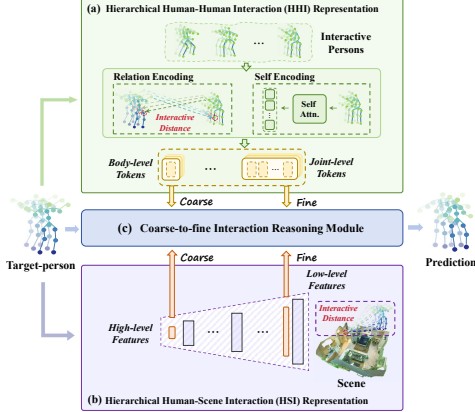

Figure 2: HUMOF Overview.

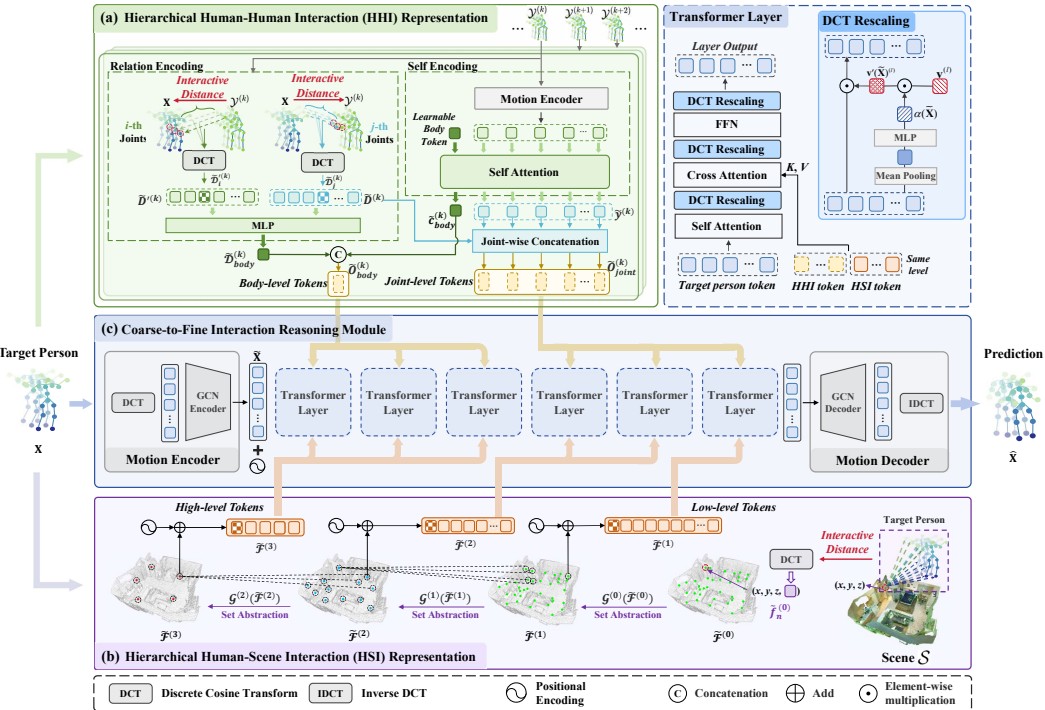

Figure 3: Detailed architecture of HUMOF. Our method takes inputs from three aspects: the past motions of the target person, a 3D point cloud for the scene, and motion sequences of interactive persons. The interactions are comprehensively encoded by (a) Hierarchical Human-Human Interaction Representation and (b) Hierarchical Human-Scene Interaction Representation, respectively. Thereafter, the hierarchical representations are leveraged by (c), a Coarse-to-Fine Interaction Reasoning Module, to predict future motions for the target person. Details of the Interaction-Perceptive Transformer layer in (c) are shown on the top right.

of $J$ body joints, each with $H$ consecutive poses. Similarly, $\mathbf{y}_j^{(k)} = \{\mathbf{y}_j^{(k)1}, \cdots, \mathbf{y}_j^{(k)H}\} \in \mathbb{R}^{H \times 3}$ represents the motion of $j$-th joint of the $k_{th}$ interactive person, with each $\mathbf{y}_j^{(k)t}$ corresponding to the 3D coordinates of one of his joint at $t_{th}$ frame. **Our goal** is to predict the motion $\mathbf{X}^{H+1:H+T}$ of the target person for the future $T$ time steps.

## 3.1 MOTION ENCODER

Follow prior works (Mao et al., 2022; Scofano et al., 2023; Xiao et al., 2025; Xing et al., 2025), we first pad the sequence $\mathbf{X}^{1:H}$ of length $H$ by repeating the last historical pose $\mathbf{X}^H$ for $T$ additional frames, to make a padded sequence of length $H + T$. For simplicity, we still call the padded sequence $\mathbf{X}$. Providing that DCT is effective in handling temporal information in human motion prediction (Mao et al., 2022; Scofano et al., 2023; Xing et al., 2025; Xiao et al., 2025), and that GCN excels at uncovering spatial dependencies between human joints (Mao et al., 2019; Xing et al., 2025; Li et al., 2020; 2022), we combine a Discrete Cosine Transform layer (DCT) and a Graph Convolutional Network (GCN) (Mao et al., 2019) to extract both spatial and temporal representations in the motion encoder (Figure 3(c)left). To help the model identify different joints, a learnable position embedding $\mathcal{P} \in \mathbb{R}^{J \times C'}$ is added to each joint. Here $C' = C \times 3$, where $C = 20$ is the number of DCT coefficients and 3 corresponds to the three directions: $x$, $y$, and $z$. Finally, the encoding $\tilde{\mathbf{X}} \in \mathbb{R}^{J \times C'}$ for a person can be formulated as

$$\tilde{\mathbf{X}} = \mathrm{GCN}(\mathrm{DCT}(\mathbf{X})) + \mathcal{P}, \tag{1}$$

which encodes features in the frequency domain for each joint $\tilde{\mathbf{x}}_j$ over the entire motion sequence.

## 3.2 HIERARCHICAL INTERACTION REPRESENTATION

Complex dynamic scenes involve interactions between humans (Section. 3.2.1), as well as between humans and their environment (Section. 3.2.2). To achieve a comprehensive representation of both human-human and human-scene interactions, we incorporate hierarchical features so that high-level features capture the overall context of the interactions, while low-level features focus on fine-grained details. This multi-level approach ensures a thorough and nuanced understanding of the interactions.

### 3.2.1 HIERARCHICAL REPRESENTATION FOR HUMAN-HUMAN INTERACTION

Regarding human-level interactions (Figure 3a), when a person engages with others, they are involved in two types of motion: independent motion, such as walking, and interactive motion, such as approaching a person to converse or adjusting one's path to avoid a collision. Therefore, we introduce a self-encoding submodule (Figure 3a right) to describe their independent motions and a relation-encoding submodule (Figure 3a left) to model their interdependencies.

**Self Encoding.** For each interactive person, we encode their motion sequence independently, capturing semantic information specific to his motion, as shown in Figure 3(a) right. This self-encoding step enables each person's motion to contribute meaningful social cues. Specifically, the motion sequence $\mathcal{Y}^{(k)}$ of $k_{th}$ interactive person is first processed through a motion encoder as described in Section 3.1, obtaining **joint-level features** in the frequency domain $\tilde{\mathcal{Y}}^{(k)} = \{\tilde{\mathbf{y}}_1^{(k)}, \cdots, \tilde{\mathbf{y}}_J^{(k)}\} \in \mathbb{R}^{J \times C'}$. Then a two-layer Transformer processes a learnable body-level feature $\tilde{c}_{body}^{(k)}$ together with the joint-level feature $\tilde{\mathcal{Y}}^{(k)}$. Note that **body-level feature** $\tilde{c}_{body}^{(k)}$ contains the information from all joints after being passed by the Transformer, serving as the body-level self encoding. While the updated joint tokens $\tilde{\mathcal{Y}}^{(k)} = \{\tilde{\mathbf{y}}_1^{(k)}, \cdots, \tilde{\mathbf{y}}_J^{(k)}\}$ constitute the joint-level self encoding.

**Relation Encoding.** We observe that despite of the various types of interactions, different interactions always lead to distinct distance patterns. Therefore, it is effective and efficient to model interactions with "distances". Hence we model the interactions explicitly to capture their dependencies via defining interactive distances as shown in Figure 3(a) left. First, for the $j_{th}$ joint of the $k_{th}$ interactive person, we calculate the interactive distance between this joint and the closest joint of the target person for each of the $H$ frames as the **joint-level relation encoding**. Specifically, at $t_{th}$ frame, the joint-level interactive distance $\mathbf{D}_j^{(k)t}$ is computed as:

$$\mathbf{D}_j^{(k)t} = \phi(\min_{i \in [1,J]} \left\| \mathbf{y}_j^{(k)t} - \mathbf{x}_i^t \right\|_2^2),\tag{2}$$

where $\phi(\cdot)$ is a mapping function such that closer joints have higher values than more distant ones. Then, we convert the time series of interactive distances $\{\mathbf{D}_j^{(k)1}, \cdots, \mathbf{D}_j^{(k)H}\}$ into frequency domain via DCT to get the joint-level relationship encoding $\tilde{\mathbf{D}}_j^{(k)} \in \mathbb{R}^C$. Second, for the $i_{th}$ joint of the target person, we similarly calculate its joint-level relationship encoding with the $k_{th}$ interactive person, denoted as $\tilde{\mathbf{D}}'_i^{(k)}$. Thereafter, we obtained **the body-level relation encoding** $\tilde{\mathbf{D}}_{body}^{(k)} = \text{MLP}(\text{concat}(\tilde{\mathbf{D}}_1^{(k)}, ..., \tilde{\mathbf{D}}_J^{(k)}, \tilde{\mathbf{D}}'_1^{(k)}, ..., \tilde{\mathbf{D}}'_J^{(k)}))$.

**Human-Human Interaction Tokens.** Finally, we concatenate the Self Encoding and Relation Encoding on their respective levels to obtain the Human-Human Interaction (HHI) token. To be clear, the $k_{th}$ interactive person's body-level HHI token is $\tilde{O}_{body}^{(k)} = \text{concat}(\tilde{c}_{body}^{(k)}, \tilde{\mathbf{D}}_{body}^{(k)})$, and joint-level HHI token is $\tilde{O}_{joint}^{(k)} = \{\tilde{o}_1^{(k)}, \cdots, \tilde{o}_J^{(k)}\}$, where $\tilde{o}_j^{(k)} = \text{concat}(\tilde{\mathbf{y}}_j^{(k)}, \tilde{\mathbf{D}}_j^{(k)})$.

### 3.2.2 HIERARCHICAL REPRESENTATION FOR HUMAN-SCENE INTERACTION

Considering the vast number of points in the 3D point cloud of a scene, it is impractical and inefficient to enumerate the target person's interactions with every point. Recalling that a centre point is frequently used to represent its neighbouring points as an approximation in geometric processing, we hope to progressively approximate neighbouring points through central points, reducing the total number of points while retaining essential scene information. In this way, we can construct different levels of point approximations with a gradually decreasing number of points, ensuring to maintain

rich interaction features across different spatial scales. Meanwhile, noting that most raw 3D scene point clouds lack object-level annotations, our method does not rely on predefined semantic labels as required by SAST (Mueller et al., 2024).

As illustrated in Figure 3(b), to obtain hierarchical point approximations, we employ a series of set abstraction layers from PointNet++ (Qi et al., 2017), denoted as $\{\mathcal{G}^{(0)}, \ldots, \mathcal{G}^{(b)}\}$, $b \in [0, 3]$. At each level of abstraction, we apply Farthest Point Sampling following PointNet++ to obtain point subsets. Each set abstraction operation processes and refines the point set to create a new set with fewer points, preserving efficiency and structure within the point cloud.

Notice that the set abstraction layer $\mathcal{G}^{(0)}$ takes an interactive feature matrix $\tilde{\mathcal{F}}^{(0)} = \{\tilde{f}_1^{(0)}, \ldots, \tilde{f}_{N^{(0)}}^{(0)}\}$ as input, where the interactive feature $\tilde{f}_n^{(0)}$ of a point $s_n$ is computed as a collection of interactive distances in the frequency domain. Specifically, for a point $s_n$ and a joint $\mathbf{x}_j$, we firstly calculate their interactive distance in each frame, constituting a time series $m_j$ :

$$m_j = \{\phi(\left\|s_n - \mathbf{x}_j^1\right\|_2^2), \cdots, \phi(\left\|s_n - \mathbf{x}_j^H\right\|_2^2)\} \in \mathbb{R}^H, \tag{3}$$

where $\phi(\cdot)$ is a mapping function that closer scene points have higher values than more distant ones (Mao et al., 2022). Next, we convert $m_j$ into frequency domain and obtain $\tilde{m}_j \in \mathbb{R}^{C'}$. Finally, we concatenate $\tilde{m}_j$ from all joints, along with the coordinates of the scene point $s_n$, forming $s_n$'s interactive feature $\tilde{f}_n^{(0)} \in \mathbb{R}^{J \times C' + 3}$.

Such a feature matrix $\tilde{\mathcal{F}}^{(b)}$ is iteratively computed across subsequent set abstraction layers (Figure 3(b)), where $\tilde{\mathcal{F}}^{(b)} = \mathcal{G}^{(b-1)}(\tilde{\mathcal{F}}^{(b-1)})$. To further enhance the positional information, we add a position encoding derived from 3D spatial coordinates of each point to the corresponding feature at each abstraction level $b \in [1, 3]$. Finally, $\tilde{\mathcal{F}}^{(b)}$ serves as the Human-Scene Interaction (HSI) tokens.

### 3.3 Coarse-to-Fine Interaction Reasoning Module

Accurate human motion prediction requires capturing kinetics and dynamics, involving inherent correlations among joints, across the temporal dimension, and with the surrounding environment. To simultaneously leverage these three types of correlations, we present a coarse-to-fine interaction reasoning module. We take the target person's representation $\tilde{\mathbf{X}}$ and all the interaction features in the frequency domain including human-to-human interactions (HHI) tokens ($\tilde{O}_{body}$ and $\tilde{O}_{joint}$) and human-to-scene interactions (HSI) tokens $\tilde{\mathcal{F}}^{(b)}$ as input, the model reasons about the motion of the target person through all interaction-perceptive Transformer layers using a coarse-to-fine strategy.

#### 3.3.1 Coarse-to-Fine Injection Strategy

With the obtained hierarchical representations for interactions—both between human and human (Section 3.2.1) and between humans and scenes (Section 3.2.2)—we establish a strategy to fully leverage this information.

Different from crudely injecting features from multiple levels of the hierarchical representation into each interaction layer of the model, we sequentially inject hierarchical interaction features in a coarse-to-fine manner. We assign high-level features to early layers and progressively incorporate low-level features at deeper layers, as shown in Figure 3(c). For example, at the first layer, high-level HSI tokens $\tilde{\mathcal{F}}^{(3)}$ and HHI tokens $\tilde{O}_{body}$ are concatenated along token dimension are injected, totaling $N^{(3)} + K$ interaction tokens. At the last layer, we inject low-level HSI tokens $\tilde{\mathcal{F}}^{(1)}$ and HHI tokens $\tilde{O}_{joint}$, totaling $N^{(1)} + K \times J$ tokens. It allows the model to begin with a global understanding of high-level semantics and gradually narrow its focus to local geometry, improving prediction accuracy.

#### 3.3.2 Interaction-Perceptive Transformer Layer

As depicted in the upper right of Figure 3, our Transformer layer begins with processing the target person's joint tokens $\tilde{\mathbf{x}}_j^{(l)}$ via a self-attention (SA) designed to capture long-range dependencies among joints. To incorporate interactions, we employ a cross-attention (CA) where joint tokens $\tilde{\mathbf{x}}_j^{(l)}$ serve as queries, while interaction tokens act as keys and values. A feed forward network

(FFN) (Vaswani et al., 2017) follows CA to enhance joint tokens. More importantly, to better align with our coarse-to-fine strategy, an adaptive DCT rescaling mechanism is performed on $\tilde{\mathbf{x}}_j^{(l)}$ after each SA, CA and FFN.

**Adaptive DCT Rescaling Mechanism.** Recall that our coarse-to-fine injection strategy focuses on capturing coarse-level human motions in the early stages, it would be helpful if we could control the influence of finer details. Observing that each primary joint token is constructed from the DCT coefficients of joint motion, with each channel corresponding to a specific DCT coefficient, we introduce a learnable DCT rescaling mechanism from the frequency domain to suppress high-frequency details. This mechanism further supports our coarse-to-fine strategy by regulating the impact of high-frequency details and noise in the Transformer layer $l$ through the rescaling vector $\mathbf{v}'(\tilde{\mathbf{X}})^{(l)} \in \mathbb{R}^{C'}$, which is applied on joint tokens $\tilde{\mathbf{x}}_j^{(l)}$ after each SA, CA, and FFN in an element-wise multiplication manner, formulated as $\tilde{\mathbf{x}}_j^{(l)} \leftarrow \tilde{\mathbf{x}}_j^{(l)} \odot \mathbf{v}'(\tilde{\mathbf{X}})^{(l)}$ where

$$\mathbf{v}'(\tilde{\mathbf{X}})^{(l)} = \mathbf{v}^{(l)} \odot \boldsymbol{\alpha}(\tilde{\mathbf{X}}), \quad \boldsymbol{\alpha}(\tilde{\mathbf{X}}) = \text{MLP}(\frac{\sum_{j=1}^{J} \tilde{\mathbf{x}}_j^{(l)}}{J}) \tag{4}$$

Here, $\odot$ is element-wise multiplication, and $\mathbf{v}^{(l)}$ is a pre-defined rescaling vector shared across all joint tokens. We design $\mathbf{v}^{(l)}$ such that values are close to 1.0 for low-frequency components and progressively decrease for higher frequencies in early layers, effectively suppressing high-frequency components. The suppression is most prominent in the first layer ($l = 1$) and gradually weakens in deeper layers, with the rescaling vector eventually having all values equal to 1.0 in the last layer ($l = 6$). Moreover, for that different types of action may have varied optimal rescaling, a shared rescaling $\mathbf{v}^{(l)}$ applied uniformly across all input samples is inadequate. We thus further include a sample-adaptive vector $\boldsymbol{\alpha}(\tilde{\mathbf{X}})$ to capture such variations, which is computed by applying average pooling across all $J$ joint tokens, followed by a MLP to acquire sample-specific information.

## 3.4 Motion Decoder and Loss

As shown in Figure 3(c) right, the updated joint tokens $\tilde{\mathbf{X}}^{(6)}$ from the $6_{th}$ Transformer layer is passed into a GCN decoder and Inverse Discrete Cosine Transform (IDCT) (Mao et al., 2019) to get predicted motion sequence $\hat{\mathbf{X}}$, which is formulated by $\hat{\mathbf{X}} = \text{IDCT}(\text{GCN}(\tilde{\mathbf{X}}^{(6)})) \in \mathbb{R}^{J \times (H+T) \times 3}$.

Loss is computed as the L2 distance between the predicted path and pose and the ground-truth. Details can be found in Appendix A.

## 4 Experiments

### 4.1 Setups

**Implementation Details** cab be found in Appendix A.

**Datasets.** We conduct experiments on 2 datasets with human-human and human-scene interactions, **HIK** (Tanke et al., 2023a) and **HOI-M**[3] (Zhang et al., 2024), as well as on 2 datasets with human-scene interaction scenes, **GTA-IM** (Cao et al., 2020) and **HUMANISE** (Wang et al., 2022). Please refer to Appendix B for more dataset details.

**Baselines.** **3 Scene-Aware Methods**, ContactAware (Mao et al., 2022), GIMO (Zheng et al., 2022), STAG (Scofano et al., 2023), and MutualDistance (Xing et al., 2025). To evaluate them on dynamic datasets, we introduce the multi-person context by concatenating all persons' information to the original input of their motion decoder. **2 Social-Aware Methods**, T2P Jeong et al. (2024) and IAFormer (Xiao et al., 2025), Since they are highly dedicated to pure multi-person input, we introduce the human-scene-interaction features generated by our HUMOF. And **1 Social-Scene-Aware Method**, SAST (Mueller et al., 2024). As it requires instance segmentation of the scene as input, we provide it with the ground-truth segmentations, except on GTA-IM, where we use the segmentation predicted by (Shin et al., 2024) due to the absence of ground truth.

| Dataset | Method | Path Error (mm) | | | | | Pose Error (mm) | | | | |
|---|---|---|---|---|---|---|---|---|---|---|---|
| | | 0.5s | 1.0s | 1.5s | 2.0s | mean | 0.5s | 1.0s | 1.5s | 2.0s | mean |
| HIK | ContAware (Mao et al., 2022) | 138.3 | 251.3 | 352.4 | 430.8 | 239.3 | 87.8 | 117.1 | 136.1 | 147.8 | 106.8 |
| | GIMO (Zheng et al., 2022) | 143.0 | 259.7 | 384.2 | 487.3 | 258.6 | 85.5 | 121.6 | 142.0 | 153.0 | 109.3 |
| | STAG (Scofano et al., 2023) | 124.7 | 245.4 | 352.4 | 479.2 | 239.7 | 81.7 | 110.9 | 132.5 | 140.9 | 100.6 |
| | MutualDistance (Xing et al., 2025) | 128.7 | 253.2 | 372.5 | 479.2 | 246.0 | 82.9 | 117.2 | 138.5 | 148.2 | 105.9 |
| | T2P (Jeong et al., 2024) | 88.6 | 199.6 | 318.8 | 447.1 | 208.7 | 74.2 | 108.6 | 127.5 | 142.6 | 96.9 |
| | IAFormer (Xiao et al., 2025) | 83.9 | 195.0 | 311.1 | 434.9 | 200.1 | 71.5 | 106.5 | 125.9 | 137.7 | 95.0 |
| | SAST (Mueller et al., 2024) | 86.7 | 187.4 | 284.9 | 398.1 | 189.0 | 72.3 | 101.4 | 118.0 | 128.6 | 93.2 |
| | **Ours** | **78.8** | **177.4** | **278.8** | **388.4** | **180.7** | **71.2** | **100.6** | **116.9** | **127.1** | **90.2** |
| HOI-M³ | ContAware (Mao et al., 2022) | 125.6 | 239.9 | 285.4 | 432.9 | 236.9 | 106.2 | 152.8 | 174.3 | 197.1 | 137.5 |
| | GIMO (Zheng et al., 2022) | 131.4 | 247.7 | 300.9 | 454.4 | 255.2 | 107.9 | 155.9 | 182.6 | 207.1 | 141.0 |
| | STAG (Scofano et al., 2023) | 128.1 | 234.4 | 289.5 | 438.1 | 239.7 | 102.5 | 145.0 | 167.1 | 185.6 | 131.2 |
| | MutualDistance (Xing et al., 2025) | 83.6 | 169.7 | 278.8 | 402.8 | 189.9 | 94.4 | 137.1 | 158.2 | 181.3 | 125.3 |
| | T2P (Jeong et al., 2024) | 74.2 | 168.8 | 296.9 | 429.2 | 194.1 | 88.0 | 135.8 | 160.9 | 183.2 | 124.6 |
| | IAFormer (Xiao et al., 2025) | 69.0 | 166.6 | 290.1 | 423.5 | 186.3 | **86.1** | 135.0 | 165.9 | 180.7 | 121.6 |
| | SAST (Mueller et al., 2024) | 75.0 | 166.2 | 280.4 | 403.9 | 184.8 | 89.2 | 133.8 | 167.0 | 182.9 | 122.3 |
| | **Ours** | **67.1** | **156.6** | **268.4** | **393.1** | **174.6** | 86.3 | **129.6** | **155.0** | **172.1** | **117.9** |

Table 1: Comparisons on datasets with dynamic scenes. We compare with scene-aware methods, ContactAware, GIMO, STAG, and MutualDistance, social-aware method, IAFormer and T2P, and social-scene-aware method SAST.

| | HUMANISE Dataset Wang et al. (2022) | | | | | | | | | | | | GTA-IM Dataset Cao et al. (2020) | | | | | | | | | |
|---|---|---|---|---|---|---|---|---|---|---|---|---|---|---|---|---|---|---|---|---|---|---|
| | Seen Scenes | | | | | | Unseen Scenes | | | | | | Path Error (mm) | | | | | Pose Error (mm) | | | | |
| Method | Path Error (mm) | | | Pose Error (mm) | | | Path Error (mm) | | | Pose Error (mm) | | | | | | | | | | | | |
| | 0.5s | 1.0s | mean | 0.5s | 1.0s | mean | 0.5s | 1.0s | mean | 0.5s | 1.0s | mean | 0.5s | 1.0s | 1.5s | 2.0s | mean | 0.5s | 1.0s | 1.5s | 2.0s | mean |
| ContAware * | 52.8 | 121.1 | 57.8 | 97.6 | 141.4 | 92.9 | 53.1 | 124.0 | 58.7 | 94.0 | 139.1 | 90.3 | 44.5 | 82.6 | 125.6 | 182.9 | 87.1 | 40.1 | 54.1 | 65.2 | 77.2 | 51.8 |
| GIMO * | 70.1 | 129.2 | 72.0 | 141.4 | 150.3 | 140.2 | 77.7 | 144.0 | 80.2 | 146.3 | 159.4 | 146.5 | 52.7 | 97.8 | 160.6 | 241.7 | 110.3 | 47.9 | 60.7 | 71.1 | 82.7 | 59.9 |
| STAG * | 55.2 | 124.6 | 60.7 | 88.3 | 131.4 | 83.0 | 57.0 | 131.5 | 63.2 | 89.9 | 137.7 | 85.6 | 43.2 | 79.8 | 119.9 | 176.4 | 83.4 | 35.4 | 48.7 | 59.8 | 73.5 | 47.0 |
| SAST | 56.2 | 122.4 | 62.1 | 86.0 | 111.6 | 80.8 | 57.2 | 129.1 | 63.7 | 90.6 | 124.7 | 86.7 | 41.0 | 77.4 | 123.1 | 181.8 | 85.2 | 28.2 | 41.0 | 53.6 | 66.8 | 41.5 |
| MutualDistance * | 41.5 | 93.5 | 45.6 | 83.7 | 130.9 | 80.0 | 46.7 | 100.2 | 50.1 | 84.3 | 131.8 | 80.6 | 34.4 | 65.9 | 104.0 | 155.6 | 72.0 | 31.0 | 46.8 | 58.9 | 70.7 | 44.6 |
| **Ours** | **36.9** | **87.4** | **41.7** | **69.0** | **102.0** | **64.1** | **39.5** | **88.1** | **43.4** | **69.8** | **109.3** | **66.2** | **29.4** | **55.9** | **91.7** | **139.2** | **62.9** | **27.0** | **40.3** | **50.7** | **61.5** | **38.7** |

Table 2: Comparisons on datasets with static scenes. Results with * are from MutualDitance.

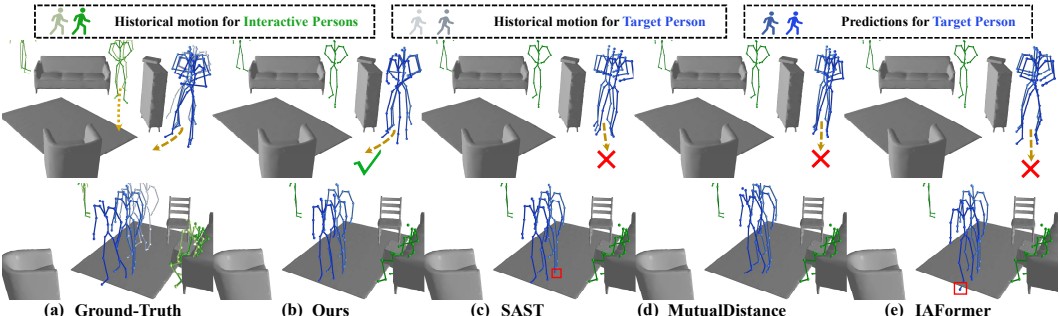

(a) Ground-Truth  (b) Ours  (c) SAST  (d) MutualDistance  (e) IAFormer

Figure 4: Visualization of motion prediction results on dynamic scenes in HOI-M³. More visual results are in the Supplementary Video and Appendix Section F.

**Metrics.** Following prior works (Mao et al., 2022; Scofano et al., 2023; Xing et al., 2025), we evaluate all methods using path error and pose error, which are computed in the same way as $\ell_{path}$ and $\ell_{pose}$ defined in Eq. 5.

## 4.2 COMPARISONS

**Evaluations on Human-human and Human-scene Interaction Scenes** We first quantitatively evaluate our approach on two real-world datasets with dynamic social scenes, HIK (Tanke et al., 2023a) and HOI-M³ (Zhang et al., 2024). As demonstrated in Table 1, our approach achieves outstanding superior performance to all other methods in three categories, highlighting our strong capability in forecasting human motion in real scenarios with complex human-human and human-scene dynamics. Visual comparisons on dynamic scenes in HOI-M³ are shown in Figure 4. In the first example, the target person exhibits a tendency to turn toward the interactive person. Only our method captures this intent and correctly predicts the direction. In the second example, SAST and IAFormer mistakenly infer some poses to be underneath the floor (marked in red boxes). In contrast, our result shows the best physical plausibility and provides the most accurate path and pose prediction. *Evaluations on long-term predictions can be found in Appendix H.*

**Evaluations on Human-Scene Interaction Scenes** are shown in Table. 2. We demonstrate significantly superior performance over SOTA scene-aware methods (Mao et al., 2022; Scofano et al., 2023; Xing et al., 2025; Zheng et al., 2022) on both the HUMANISE (Wang et al., 2022) and GTA-IM (Cao et al., 2020) datasets. Meanwhile, our method also greatly outperforms the social-scene-aware method SAST (Mueller et al., 2024). Note that we do not rely on ground truth instance segmentation, which is required by SAST.

## 4.3  DISCUSSIONS

**Joint Multi-Person Inference** Our method is readily scalable to joint multi-person motion forecasting by treating each individual as a separate target in a data sample and then inferring the batch (batch size $= 1 + K$). A visual result is shown in Figure 5.

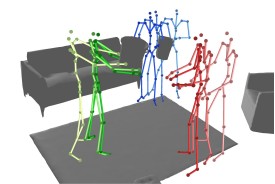

Figure 5: Joint forecasting.

**Handling of Dynamic Scene Elements** Currently, few datasets contain a significant number of dynamic scene elements. But note that our model architecture can natively handle dynamic scene elements without any structural changes. Our HSI module computes a time series of distances between the target person and scene points. If a scene point were dynamic, its coordinates would become time-dependent ($p_s \rightarrow p_s(t)$). The distance calculation naturally extends to this dynamic case, and no architectural modifications would be required. Therefore, given a dataset with significant dynamic objects, our framework could be trained to handle such scenarios directly. We provide a preliminary validation on a subset of dynamic scenes in Appendix M, showing the potential ability to handle dynamic furniture objects despite limited data.

**Scalability and Generality** Our framework is designed with flexibility and scalability. The core principle—encoding environmental information into a unified hierarchical representation and processing it with a coarse-to-fine mechanism—is not fundamentally tied to motion and point cloud data. To incorporate new modalities, such as video or audio streams, one could employ standard pre-trained encoders (e.g., ViT for images) to produce hierarchical feature tokens. These new tokens could then be seamlessly concatenated with our existing HSI and HHI tokens and processed by the interaction reasoning module. This adaptability allows the framework to be extended to a wider range of scenarios and input types without requiring a significant redesign, pointing a path toward more general and scalable models for motion forecasting.

**More Discussions on Design Choices** are included in Appendix E, covering interpretability of cross-attention weights, and choice of holistic point cloud rather than object-level scene representation.

**Runtime Analysis** Our model takes similar or shorter time to perform inference than other methods. Specifically, our model has 9.6M parameters and achieves an inference time of 43ms on HOI-M$^3$. See Appendix Section C for more detailed comparisons with baselines.

## 4.4  ABLATION STUDIES ON THE HOI-M$^3$ DATASET

All ablated variants share the same network depth, and the parameter counts of all variants are basically the same, ranging from 9M to 11M. More ablations on *variants of Coarse-to-Fine Interaction, different number of sampled points, different number of transformer layers, robustness to incomplete interaction information can be found in Appendix Section D.*

**Hierarchical Interaction Representation.** Table 3(a) shows the effectiveness of our human-to-scene and human-to-human (including self-encoding and relation-encoding) interaction representations. Results indicate a decline in our motion prediction performance when any of these modules are removed, underscoring their individual contributions.

**Coarse-to-Fine Interaction Feature Injection.** We assess the effectiveness of the coarse-to-fine interaction feature injection strategy in Table 3(b). The experiment shows that multi-level interaction feature injection outperforms single-level approaches (coarse-only and fine-only). But our coarse-to-fine strategy utilizes multi-level features in a more effective way and further boosts the performance.

**Adaptive DCT Rescaling.** Table 3(c) validates the effectiveness of the adaptive DCT rescaling mechanism. By suppressing high-frequency updates of joint motion in early stages to encourage the

focus on coarse and low-frequency updates, the shared static rescaling vector $\mathbf{v}^{(l)}$ itself improves prediction accuracy. Besides, the performance is further enhanced after combining the sample-adaptive vector $\boldsymbol{\alpha}(\tilde{\mathbf{X}})$, as it allows the frequency rescalings to adapt to different input samples.

(a) Hierarchical Representations

| HSI | HHI | | Path Error (mm) | | | Pose Error (mm) | | |
|---|---|---|---|---|---|---|---|---|
| | Self | Relation | 1.0s | 2.0s | mean | 1.0s | 2.0s | mean |
| ✗ | ✗ | ✗ | 167.0 | 426.8 | 187.6 | 134.7 | 181.8 | 123.2 |
| ✗ | ✓ | ✓ | 164.1 | 415.7 | 183.7 | 133.2 | 175.5 | 120.9 |
| ✓ | ✗ | ✗ | 163.0 | 414.0 | 182.9 | 132.8 | 179.2 | 121.4 |
| ✓ | ✓ | ✗ | 160.0 | 402.7 | 178.4 | 132.0 | 175.0 | 120.0 |
| ✓ | ✗ | ✓ | 158.2 | 401.0 | 177.0 | 131.4 | 175.6 | 119.9 |
| ✓ | ✓ | ✓ | 156.6 | 393.1 | 174.6 | 129.6 | 172.1 | 117.9 |

(b) Injection Strategy

| Injection strategy | Path Error (mm) | | | Pose Error (mm) | | |
|---|---|---|---|---|---|---|
| | 1.0s | 2.0s | mean | 1.0s | 2.0s | mean |
| Coarse-only | 164.6 | 411.0 | 182.7 | 133.8 | 175.3 | 121.2 |
| Fine-only | 163.5 | 409.5 | 182.4 | 133.3 | 178.9 | 121.2 |
| Multi-level | 158.8 | 398.9 | 177.1 | 133.5 | 174.7 | 120.5 |
| Coarse-to-Fine (Ours) | 156.6 | 393.1 | 174.6 | 129.6 | 172.1 | 117.9 |

(c) Adaptive DCT Rescaling

| DCT rescaling | | Path Error (mm) | | | Pose Error (mm) | | |
|---|---|---|---|---|---|---|---|
| $\mathbf{v}^{(l)}$ | $\boldsymbol{\alpha}(\tilde{\mathbf{X}})$ | 1.0s | 2.0s | mean | 1.0s | 2.0s | mean |
| ✗ | ✗ | 159.6 | 404.3 | 178.9 | 132.0 | 173.9 | 120.2 |
| ✓ | ✗ | 157.2 | 398.4 | 176.3 | 130.6 | 174.4 | 119.2 |
| ✗ | ✓ | 158.0 | 401.0 | 177.2 | 131.7 | 174.6 | 119.7 |
| ✓ | ✓ | 156.6 | 393.1 | 174.6 | 129.6 | 172.1 | 117.9 |

Table 3: Ablations studies.

## 5 CONCLUSIONS

In conclusion, we present an effective approach for human motion forecasting in interactive environments. By representing hierarchical interactive features and employing the coarse-to-fine interaction reasoning module, our method achieves state-of-the-art performance across four public datasets, demonstrating the potential to construct a world model for human motion. This approach holds significant promise for real-world applications, such as enhancing closed-loop simulations for autonomous driving and improving the understanding and interaction capabilities of robots.

### ETHICS STATEMENT

Our work presents no direct ethical concerns. The primary application of our method is for human motion prediction.

### REPRODUCIBILITY STATEMENT

To ensure reproducibility: (1) While not included with this submission, our full project will be released on GitHub upon publication. (2) All experimental details are included in Appendix A, and (3) Usage of datasets is explained in Appendix B.

### ACKNOWLEDGEMENTS

This work was supported by MoE Key Laboratory of Intelligent Perception and Human-Machine Collaboration (KLIP-HuMaCo), Shanghai Frontiers Science Center of Human-centered Artificial Intelligence (ShangHAI), HKRGC Theme-based research scheme project T35-710/20-R, and SZ-HK-Macau Technology Research Programme #SGDX20210823103537030.

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

## APPENDIX

We organize our appendix as follows.

- In Section A, we provide more implementation details.
- In Section B, we provide more dataset details.
- In Section C, we present the runtime analysis.
- In Section D, we present more ablation results.
- In Section E, we provide more discussions on our model design choices.
- In Section F, we present more visual comparisons on GTA-IM Cao et al. (2020) dataset and HOI-M[3] Zhang et al. (2024) dataset.

- In Section G, we show performance under closer interactions.

- In Section H, we present extended evaluation with NPSS metric and long-term prediction.

- In Section I, we present extended evaluation with FID metric.

- In Section J, we report human-object and human-human penetration.

- In Section K, we discuss the limitations and future works.

- In Section L, we investigate SE(3) representations.

- In Section M, we present preliminary validation on scenes with dynamic objects.

## A  IMPLEMENTATION DETAILS

For each motion sequence, we crop the 3D scene to a region that is within 2.5 meters of the root joint of the last observed pose, and the root joint is used as the origin of the cropped scene, following prior works (Mao et al., 2022; Scofano et al., 2023). Then we obtain $\mathcal{S} \in \mathbb{R}^{N \times 3}$ by randomly sampling $N = 1000$ points in the cropped scene. The mapping function $\phi(\cdot)$ is defined as $\phi(\cdot) : d \rightarrow e^{-\frac{d^2}{2\sigma^2}}$ where $\sigma = 0.2$.

**Network Architecture.** The model consists of 6 Transformer layers with hierarchical dimensions: early layers (1-3) use dimension 512, while late layers use dimension 256 (layers 4-5) or dimension 128 (layer 6). The feed-forward dimension is 4 times the hidden dimension. All Transformer layers use 4 self-attention heads and 4 cross-attention heads. The Graph Convolutional Network (GCN) encoder uses 5 residual stages. A learnable position encoding is applied to joint tokens with dimension $3 \times C = 60$. Object nodes are augmented with 8-dimensional position encoding: 3D center coordinates, exponential encoding of center ($e^{-\frac{\|\mathbf{c}\|^2}{2\sigma^2}}$), distance to origin, and exponential encoding of distance. Object point clouds are normalized by subtracting their center.

**Loss Function.** Loss is computed as $\ell = \ell_{\text{path}} + \ell_{\text{pose}}$. The path loss $\ell_{\text{path}}$ and local pose loss $\ell_{\text{local}}$ are defined as

$$
\begin{aligned}
\ell_{\text{path}} &= \frac{1}{T} \sum_{t=H+1}^{H+T} \left\| \mathbf{X}_{\text{root}}^t - \hat{\mathbf{X}}_{\text{root}}^t \right\|_2^2 \\
\ell_{\text{local}} &= \frac{1}{T(J-1)} \sum_{t=H+1}^{H+T} \sum_{j=1}^{J-1} \left\| \mathbf{X}_{\text{local},j}^t - \hat{\mathbf{X}}_{\text{local},j}^t \right\|_2^2 .
\end{aligned}
\tag{5}
$$

Here, $\mathbf{X}_{\text{root}}^t \in \mathbb{R}^3$ and $\hat{\mathbf{X}}_{\text{root}}^t \in \mathbb{R}^3$ are ground-truth and the predicted global path of the root joint at time $t$. $\mathbf{X}_{\text{local},j}^t \in \mathbb{R}^3$ and $\hat{\mathbf{X}}_{\text{local},j}^t \in \mathbb{R}^3$ are ground-truth and predicted local pose of the $j^{th}$ non-root joint at time $t$.

**More Training Details.** We build our network on PyTorch 1.12.0 and CUDA 12.4. We follow previous work Mao et al. (2022) to use the Adam optimizer with a linear learning rate schedule from 0.0005 to 0. The initial learning rate is 0.0005. Models are trained over 80 epochs. Weight decay is set to $1 \times 10^{-6}$, and Adam epsilon is $\epsilon = 1 \times 10^{-6}$. We use a dropout of 0.1. For single-person datasets, the batch size is 16. For multi-person datasets, dynamic batch sampling is used with a maximum sum of other persons set to 256 (in one batch, the number of other persons for each sample should be the same. Thus we use a custom batch sampler such that in each batch sample, the number of other individuals is the same across each sample). Rotation augmentation is applied during training so that the model can learn the direction-agnostic representation of the inputs.

**More Dataset Details.** Following prior works Mao et al. (2022); Scofano et al. (2023); Xing et al. (2025); Mueller et al. (2024), the FPS of four datasets are: GTA-IM: 30 Mao et al. (2022); Scofano et al. (2023); Xing et al. (2025), HUMANISE: 30 Xing et al. (2025), HOI-M$^3$: 30, and HIK: 25 Mueller et al. (2024). We use $H = 30$ motion frames to predict $T = 60$ future steps for datasets HOI-M$^3$ (Zhang et al., 2024) and GTA-IM (Cao et al., 2020), $H = 25$ and $T = 50$ for HIK (Tanke et al., 2023a), $H = 15$ and $T = 30$ for HUMANISE (Wang et al., 2022).

# B DATASET

**A. Datasets with Human-human and Human-scene Interaction Scenes.** 1) **HIK** (Tanke et al., 2023a) is a multi-person interaction datasets in real kitchen environments. We follow the dataset split used in (Tanke et al., 2023a; Mueller et al., 2024), using the recordings A-C as training data and evaluating on the recording D. 2) **HOI-M**$^3$ (Zhang et al., 2024) captures a rich collection of interactions involving multiple humans and objects across 46 diverse scenes in the real world. We randomly allocate 1/5 of these scenes for the test set and utilize the remaining for training. For a datasets, we filter out sequences with few social interactions and human movements, and retain those with significant interactions and motion displacement.

**B. Datasets with Human-Scene Interaction Scenes.** 1) **GTA-IM** (Cao et al., 2020) is a synthetic human-scene interaction dataset, comprising 3D human motions for 50 distinct characters across 7 diverse scenes. We adopt the same dataset setting as Mao et al. (2022); Scofano et al. (2023); Xing et al. (2025). 2) **HUMANISE** (Wang et al., 2022) is a synthetic human-scene interaction dataset.

All methods adopt the dataset setting used in MutualDistance (Xing et al., 2025) which ensures motions in the test set are entirely unseen during training. The test set scenes are further divided into seen and unseen scenes, with about 6,000 sub-sequences for testing.

# C RUNTIME ANALYSIS

An analysis of inference time and model size is shown in Table 4. Overall, our model takes a reasonable time to perform inference while achieving higher accuracy than baselines. It is worth noting that though SAST is designed to solve the problem under a similar setting as us, it is not practical for real-world applications as it adopts a diffusion mechanism, which makes it slow.

Table 4: Runtime analysis on HOI-M$^3$ Dataset.

| Method | # Param. | Inference Time |
|---|---|---|
| ContAware Mao et al. (2022) | 15.9 M | 41 ms |
| STAG Scofano et al. (2023) | 16.4 M | 38 ms |
| MutualDistance Xing et al. (2025) | 8.6 M | 114 ms |
| IAFormer Xiao et al. (2025) | 9.2 M | 69 ms |
| SAST Mueller et al. (2024) | 15.4 M | 2 s |
| **Ours** | 9.6 M | 43 ms |

# D MORE ABLATION RESULTS

## D.1 VARIANTS OF COARSE-TO-FINE INTERACTION

For the coarse-to-fine injection of interaction features at different levels, there are dozens of potential variants. In our main paper, we adopt a relatively symmetric injection strategy, where high-level HSI and HHI features are injected into the first three layers, while low-level or mid-level features are injected into the later three layers. Here, we conduct an ablation study on different variants of the injection strategy.

For simplicity, we fix the injection strategy of HHI features i.e., body-level tokens are injected into the first three layers and joint-level tokens into the last three layers. We only apply variations to the hierarchical HSI tokens $\tilde{\mathcal{F}}^{(1)}$, $\tilde{\mathcal{F}}^{(2)}$, and $\tilde{\mathcal{F}}^{(3)}$. Specifically, we evaluate four distinct injection variants as depicted in Table 5.

As shown in Table 5, the four variants exhibit similar prediction performance, indicating that the model is insensitive to the specific injection variant chosen. The prediction errors are consistently lower than those of the non-coarse-to-fine methods presented in Table 5. This experiment demonstrates the robustness and effectiveness of the coarse-to-fine interaction strategy.

| Method | Variant $\tilde{\mathcal{F}}^{(3)}$ | $\tilde{\mathcal{F}}^{(2)}$ | $\tilde{\mathcal{F}}^{(1)}$ | Mean Path Error (mm) | Mean Pose Error (mm) |
|---|---|---|---|---|---|
| Multi-level | 1∼6 | 1∼6 | 1∼6 | 177.1 | 120.5 |
| Coarse-to-Fine | 1,2,3 | 4,5 | 6 | 174.6 | 117.9 |
|  | 1,2,3 | 4 | 5,6 | 175.2 | 118.1 |
|  | 1,2 | 3,4 | 5,6 | 174.8 | 117.9 |
|  | 1,2 | 3 | 4,5,6 | 175.0 | 118.2 |

Table 5: Impact of different variants of coarse-to-fine injection. We report the metrics on the HOI-M³ Zhang et al. (2024) dataset. The *Variant* column specifies which Transformer layers receive HSI features at each level. For instance, the last row means that $\tilde{\mathcal{F}}^{(3)}$ is injected into Transformer layers 1 and 2, $\tilde{\mathcal{F}}^{(2)}$ into Transformer layer 3, and $\tilde{\mathcal{F}}^{(1)}$ into Transformer layers 4, 5, and 6. This experiment indicates that we are **insensitive** to different variants of coarse-to-fine injection, demonstrating the robustness and effectiveness of the coarse-to-fine injection.

## D.2 ABLATION STUDY ON DIFFERENT NUMBER OF SAMPLED POINTS

As shown in Table 6, the performance differences between 1000 and 4000 sampled points are negligible. However, reducing the number of points to 250 leads to a degradation in both path and pose accuracy. To balance computational efficiency with performance, we select 1000 points as the default configuration in our experiments.

| #points | Path Error (mm) | | | | | Pose Error (mm) | | | | |
|---|---|---|---|---|---|---|---|---|---|---|
|  | 0.5s | 1.0s | 1.5s | 2.0s | mean | 0.5s | 1.0s | 1.5s | 2.0s | mean |
| 250 | 29.9 | 57.2 | 94.2 | 148.7 | 65.2 | 27.8 | 42.6 | 53.9 | 64.1 | 40.5 |
| 1000(default) | 29.4 | 55.9 | 91.7 | 139.2 | 62.9 | 27.0 | 40.3 | 50.7 | 61.5 | 38.7 |
| 4000 | 29.2 | 55.9 | 91.8 | 139.7 | 63.1 | 27.0 | 40.1 | 50.3 | 60.8 | 38.4 |

Table 6: Ablation study on different number of sampled point of the static scene on GTA-IM dataset.

## D.3 ABLATION STUDY ON DIFFERENT NUMBER OF TRANSFORMER LAYERS

We ablate on the number of Transformer layers of the Coarse-to-Fine Interaction Reasoning Module. For feature injection, we adopt a symmetric strategy: high-level HSI (scene) and HHI (human-human) features are injected into the first half of the layers, while low- and mid-level features are injected into the latter half. Table 7 reveals that while deeper architectures (more layers) generally achieve better accuracy, the rate of improvement decreases as we add more layers. To balance computational efficiency with performance, we adopt 6 layers as our default configuration.

| #Transformer layers | Path Error (mm) | | | | | Pose Error (mm) | | | | |
|---|---|---|---|---|---|---|---|---|---|---|
|  | 0.5s | 1.0s | 1.5s | 2.0s | mean | 0.5s | 1.0s | 1.5s | 2.0s | mean |
| 4 | 30.3 | 58.1 | 93.3 | 140.4 | 64.0 | 28.2 | 42.2 | 52.8 | 63.7 | 40.2 |
| 6(default) | 29.4 | 55.9 | 91.7 | 139.2 | 62.9 | 27.0 | 40.3 | 50.7 | 61.5 | 38.7 |
| 8 | 28.9 | 55.4 | 90.4 | 135.2 | 61.9 | 26.3 | 39.5 | 49.9 | 61.0 | 38.3 |

Table 7: Ablation study on different number of Transformer layers on GTA-IM dataset.

## D.4 ROBUSTNESS TO INCOMPLETE INTERACTION INFORMATION

To evaluate the robustness of our model against incomplete information, we conduct experiments where parts of the scene and some interacting individuals are randomly occluded. For the scene, we

simulate occlusion by randomly removing points within 6 cones originating from the target person. For other individuals, we randomly remove 0-2 persons from the scene. As shown in Table 8, while our method's performance sees a slight degradation as expected, it still outperforms other methods, demonstrating its robustness.

Table 8: Evaluation with incomplete interaction information on the HOI-M3 dataset. We report mean Path Error (mm) and Pose Error (mm). Lower is better.

| Method | Path Error | Pose Error |
|---|---|---|
| MutualDistance | 190.7 | 125.3 |
| IAFormer | 186.4 | 121.6 |
| SAST | 185.9 | 122.7 |
| **Ours** | **176.6** | **118.1** |
| Ours (w/o occlusion) | 174.6 | 117.9 |

We further analyze the impact of varying scene occlusion levels on model performance. By increasing the number of occlusion cones from 2 to 16, we simulate progressively severe scene incompleteness. As shown in Figure 6, while prediction error naturally increases with occlusion severity, our method maintains a consistent performance advantage over baselines.

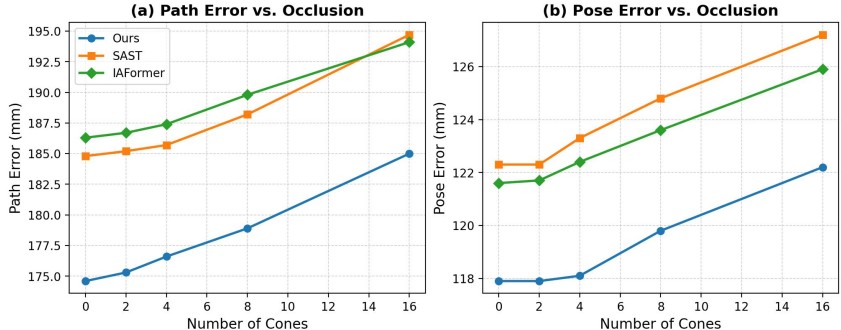

Figure 6: Performance under varying levels of scene occlusion. The x-axis represents the number of occlusion cones (indicating occlusion severity), and the y-axis represents the Mean Error.

### D.5 ROBUSTNESS TO NOISY INPUTS

We conducted experiments using noisy inputs (injecting Gaussian noise into the joints of other individuals and the scene point cloud). As illustrated in Figure 7, while performance naturally degrades for all methods as noise levels increase, HUMOF consistently outperforms the baselines. This demonstrates that our approach remains effective and robust even with imperfect inputs.

## E FURTHER DISCUSSIONS ON MODEL DESIGN AND CAPABILITIES

### E.1 RATIONALE FOR NOT PREDICTING A SPECIFIC INTERACTIVE TARGET

Our decision not to build in an inductive bias for predicting an explicit interaction partner is a deliberate design choice, motivated by the complex and fluid nature of real-world human interactions. In the real world, human motion is rarely governed by a singular environment element. Instead, it often results from a blend of multiple surrounding entities. For instance, a person might navigate around a table while simultaneously turning their head to speak to a friend. The final motion is a synthesis of these concurrent spatial and social cues. Therefore, forcing the model to explicitly predict a specific interaction target would be an ill-posed simplification, failing to capture the rich, blended nature of these influences.

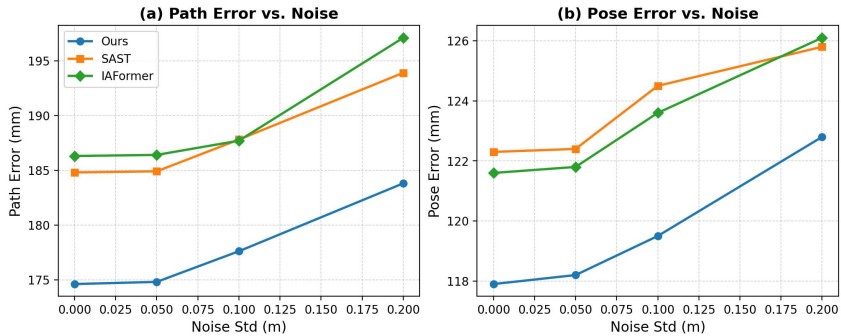

Figure 7: Robustness analysis under noisy inputs. The plot shows performance degradation with increasing Gaussian noise.

Instead, we build interaction features for all nearby individuals and scene points and allow a cross-attention mechanism to dynamically weigh the influence of each. This approach is more flexible and better reflects the complex nature of social dynamics.

### E.1.1  INTERPRETABILITY OF CROSS-ATTENTION WEIGHTS

Our decision not to explicitly predict a specific interaction target does not mean that our model is a black box. The cross-attention mechanism in our framework offers a way to interpret the model's focus. To demonstrate this, we provide a quantitative analysis of the attention weights. On the Humanise dataset, which provides ground-truth labels for the primary object a person interacts with, we find that the HSI tokens with the highest attention scores often correspond to that ground-truth object. Specifically, we identify the top-3 HSI tokens with the highest average attention scores for each sample and check if any of them correspond to the ground-truth interaction object. We formalize this as:

$$\text{Accuracy} = \frac{\sum_1^N \mathbb{I}\left(T_{\text{top-3}} \text{ corresponds to } O_{\text{GT}}\right)}{N} = 83.85\%$$

$$\text{Baseline (Random Chance) Accuracy} = \frac{\sum_1^N \frac{3}{\text{num of obj}}}{N} = 10.56\%$$

As shown in Table 9, our attention-based identification achieves an accuracy of 83.85%, significantly outperforming a random-chance baseline. This result quantitatively validates that our model learns to focus on relevant objects and is not simply overfitting, providing interpretable insights into its decision-making process.

Table 9: Interpretability of the cross-attention mechanism. We report Top-3 accuracy for identifying the ground-truth interactive object on the Humanise dataset. Higher is better.

| Method | Top-3 Accuracy |
|---|---|
| Random Chance | 10.56% |
| Via Attention Map of Our Model | **83.85%** |

### E.2  SCENE REPRESENTATION: HOLISTIC POINT CLOUD VS. OBJECT-LEVEL MODELING

In our framework, we model the scene as a holistic point cloud rather than segmenting it into individual objects. This design choice is motivated by two primary factors: efficiency and practical applicability. Modeling every object individually, especially in complex scenes with numerous objects, would introduce significant computational overhead. More importantly, it would create a dependency on accurate and readily available instance segmentation, which is often not the case in real-world scenarios that rely on raw sensor data. Our approach avoids this dependency. The strong

performance of our method across multiple datasets validates the effectiveness of this modeling strategy.

## F   MORE VISUAL COMPARISONS

We provide additional visualization results on the GTA-IM Cao et al. (2020) dataset in Figure 8 and the HOI-M[3] dataset in Figure 9. Our method demonstrates superior accuracy in predicting human motion, including both global trajectories and local poses.

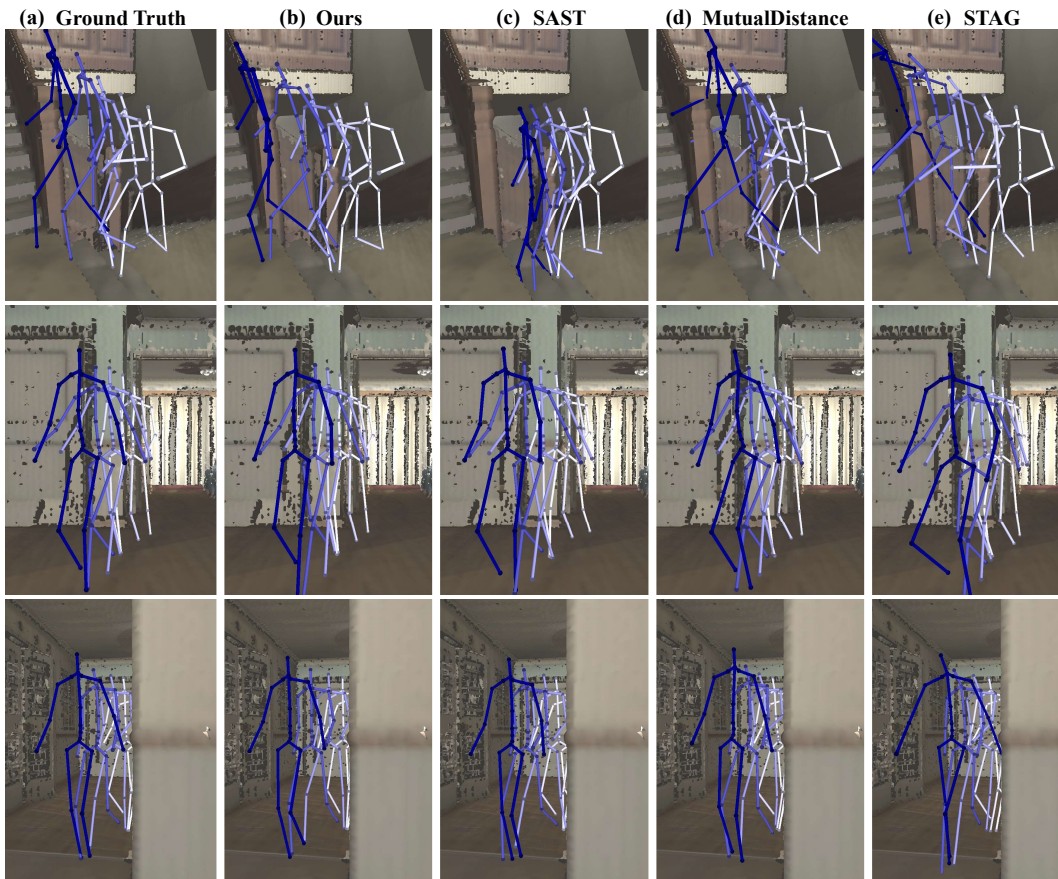

Figure 8: Visual comparisons on the GTA-IM Cao et al. (2020) dataset. Our method produces the best predictions. For instance, in the 1st row, SAST Mueller et al. (2024) predicts poses that intersect with the space beneath the stairs, likely due to its lack of explicit modeling of human-scene interactions. While MutualDistance Xing et al. (2025) and STAG Scofano et al. (2023) avoid this issue, they also produce inaccurate predictions. Our method generates predictions closest to the ground truth.

## G   PERFORMANCE UNDER CLOSER INTERACTIONS

To evaluate the model's capability in handling intense interactions, we conducted an evaluation on a specific subset of the HOI-M[3] dataset focusing on **close interactions**, where the target person is in close proximity ($< 15$cm) to scene objects or other individuals.

As shown in Table 10, the DCT-based method does not limit interaction quality. The results on the subset are even slightly better compared to the average result on the full test set. This is likely because when there are closer scene objects or other individuals, they provide stronger geometric constraints on the target motion. Thus, it becomes easier for the model to give a better prediction compared to samples where there are no very close scene objects or other persons.

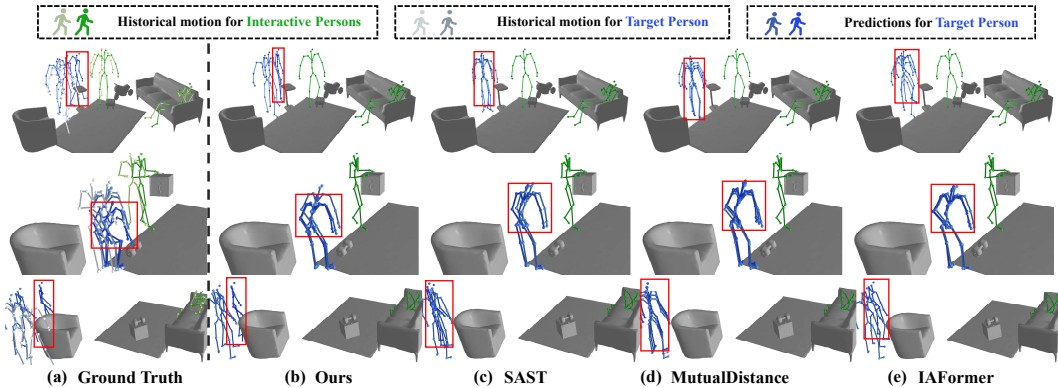

Figure 9: More visual comparisons on HOI-M$^3$ dataset.

Table 10: Comparison of performance between the full test set and the close-interaction subset on the HOI-M$^3$ dataset.

| Test Set | Mean Path Error (mm) ↓ | Mean Pose Error (mm) ↓ |
|---|---|---|
| Ours (Full Test Set) | 174.6 | 117.9 |
| Ours (Close Interaction Subset) | 172.5 | 116.8 |

We also provide visual examples of these close human-human (InterX dataset Xu et al. (2024)) and human-scene (HUMANISE dataset) interactions in the Supplementary Video (1:23 to 1:59) to further demonstrate the high interaction quality.

## H    EXTENDED EVALUATION: NPSS METRIC AND LONG-TERM PREDICTION

In the main paper, we follow prior works Mao et al. (2022); Scofano et al. (2023); Xing et al. (2025) and report Path Error and Pose Error as metrics. However, as noted in Gopalakrishnan et al. (2019), for prediction horizons longer than one second, the inherent stochasticity of human motion can make traditional geometric error metrics less informative. To provide a more comprehensive evaluation, we additionally add the normalized power spectrum similarity (NPSS) metric, which offers a statistical evaluation of motion quality.

The NPSS is calculated as the Euclidean distance between the power spectra of the prediction and the ground truth. The formula is:

$$\text{NPSS} = \frac{1}{D} \sum_{d=1}^{D} \sqrt{\sum_{k=1}^{C} (P_{d,k} - \hat{P}_{d,k})^2}$$

where $D$ is the number of dimensions (joints × coordinates), $C$ is the number of DCT coefficients, $P$ is the power spectrum of the ground-truth sequence, and $\hat{P}$ corresponds to the predicted sequence Gopalakrishnan et al. (2019).

While our primary focus is on short-term prediction (up to 2 seconds), we also evaluated our method on a challenging long-term prediction task (10 seconds) on the HOI-M$^3$ dataset to provide a reference. As shown in Table 12, the performance of all methods degrades significantly as the prediction horizon increases, which is expected due to the compounding uncertainty in long-term forecasting. Nevertheless, our method consistently outperforms the baselines across all metrics, demonstrating its robustness for longer-term predictions.

Table 11: Quantitative results with the NPSS metric for 2-second prediction on the HIK and HOI-M$^3$ datasets. This table extends Table 1 from the main paper by including the NPSS metric.

| Dataset | Method | Mean Path Error (mm) ↓ | Mean Pose Error (mm) ↓ | NPSS ↓ |
|---|---|---|---|---|
| HIK | MutualDistance Xing et al. (2025) | 246.0 | 105.9 | 0.00778 |
| | IAFormer Xiao et al. (2025) | 200.1 | 95.0 | 0.00718 |
| | SAST Mueller et al. (2024) | 189.0 | 93.2 | 0.00711 |
| | **Ours** | **180.7** | **90.2** | **0.00703** |
| HOI-M$^3$ | MutualDistance Xing et al. (2025) | 189.9 | 125.3 | 0.0195 |
| | IAFormer Xiao et al. (2025) | 186.3 | 121.6 | 0.0172 |
| | SAST Mueller et al. (2024) | 184.8 | 122.3 | 0.0179 |
| | **Ours** | **174.6** | **117.9** | **0.0169** |

Table 12: Long-term (10s) prediction performance on the HOI-M$^3$ dataset.

| Method | Mean Path Error (mm) ↓ | Mean Pose Error (mm) ↓ | NPSS ↓ |
|---|---|---|---|
| MutualDistance Xing et al. (2025) | 824.5 | 191.9 | 0.308 |
| IAFormer Xiao et al. (2025) | 768.3 | 203.1 | 0.246 |
| SAST Mueller et al. (2024) | 814.7 | 191.3 | 0.320 |
| **Ours** | **747.8** | **188.7** | **0.227** |

## I   MORE EVALUATION METRIC: FID

To further evaluate the quality of the generated motions on a distribution level, we calculate the Fréchet Inception Distance (FID). We utilized the pre-trained motion encoder from T2M Guo et al. (2022a) as the feature extractor to map motion sequences into a 512-dimensional feature space and then calculate FID on the HOI-M$^3$ dataset. The results are summarized in Table 13. Our method achieves the lowest FID score, demonstrating that our generated motions match the ground-truth distribution better than the baselines.

Table 13: FID scores on HOI-M$^3$ dataset.

| Method | FID (↓) |
|---|---|
| SAST Mueller et al. (2024) | 0.0278 |
| MutualDistance Xing et al. (2025) | 0.0233 |
| IAFormer Xiao et al. (2025) | 0.0170 |
| **Ours** | **0.0164** |

## J   HUMAN-OBJECT AND HUMAN-HUMAN PENETRATION

To measure human-object and human-human penetration, we calculated the mean penetration rate and penetration depth on the HOI-M$^3$ dataset, which provides the high-quality scene meshes and human body models (SMPL-X) necessary for this analysis. The metrics are defined as follows:

- **Penetration depth** at the $t$-th frame (in meters) is defined as the sum of absolute signed distance field (SDF) values for all joints of the target person that penetrate the scene or other persons:

$$\sum_{j=1}^{J} \left| \left( \Psi \left( \mathbf{X}_j^t \right) \right)_{-} \right|$$

where $\Psi(\cdot)$ denotes the signed distance field (SDF) of the scene or interactive persons, $(\cdot)_{-}$ clips all positive distances to zero, and $\mathbf{X}_j^t$ is the 3D position of the $j$-th joint at time $t$.

- **Penetration rate** at frame $t$ is the ratio of joints with a negative SDF value to the total number of joints:

$$\frac{\text{Number of joints with } \Psi \left( \mathbf{X}_j^t \right) < 0}{J}$$

We then take the average over the frame and sample dimensions to obtain the final mean metrics.

Table 14: Penetration Results on the HOI-M$^3$ dataset.

| Method | Human-to-Scene | | Human-to-Human | |
|---|---|---|---|---|
| | Mean Pen. Rate | Mean Pen. Depth (mm) | Mean Pen. Rate | Mean Pen. Depth (mm) |
| Ground-Truth (GT) | 1.84% | 11.26 | 0.070% | 0.61 |
| SAST | 1.54% | 11.07 | 0.063% | 0.45 |
| IAFormer | 1.62% | 10.89 | 0.068% | 0.59 |
| MutualDistance | **1.45%** | **10.68** | 0.057% | 0.41 |
| **Ours** | 1.49% | 10.77 | **0.052%** | **0.39** |

As shown in Table 14, MutualDistance achieves the best HSI penetration scores, likely due to its use of mesh-based modeling for both the target person and the scene, providing more explicit surface information to avoid penetration. However, we note that penetration metrics alone, without considering motion accuracy, should be interpreted with caution. Our case analysis reveals that higher penetration rates can sometimes result from accurately predicting dynamic motion, whereas lower penetration may occur when a method predicts static or incorrect motion (e.g., the person remaining stationary). This is also supported by the fact that the Ground-Truth (GT) motion itself registers the highest penetration, due to small misalignments between the motion capture data and scanned scene geometry in the dataset, whereas the four prediction methods may inadvertently avoid penetration by under-predicting movement (e.g. remaining static). Our method strikes a strong balance, achieving high motion accuracy while maintaining penetration scores comparable to or better than most baselines.

To further reduce penetration while preserving accuracy, we could also adopt mesh-based modeling—e.g., after computing joint-to-point/joint distance, we then adjust these by subtracting the point-to-mesh surface distance to get mesh-to-mesh distances. In our current method we did not adopt this design as this would introduce dependencies on scene meshes and SMPL-X parameters, increasing complexity and reducing practical applicability.

## K  LIMITATIONS AND FUTURE WORKS

**Failure cases**  Our method occasionally struggles with accurately predicting abrupt motion changes, as illustrated in Figure 10. In the upper example in Figure 10, our method fails to predict the bending-over action, as it is difficult to infer from the past motion. In the lower example, our method incorrectly predicts that the target person will continue standing by the desk, while the person unexpectedly starts walking away. Note that all methods fail in these challenging cases. These issues could potentially be mitigated by incorporating additional modalities, such as human gaze Zheng et al. (2022), to provide richer contextual information. We show the failure cases in the HIK dataset in the supplementary video.

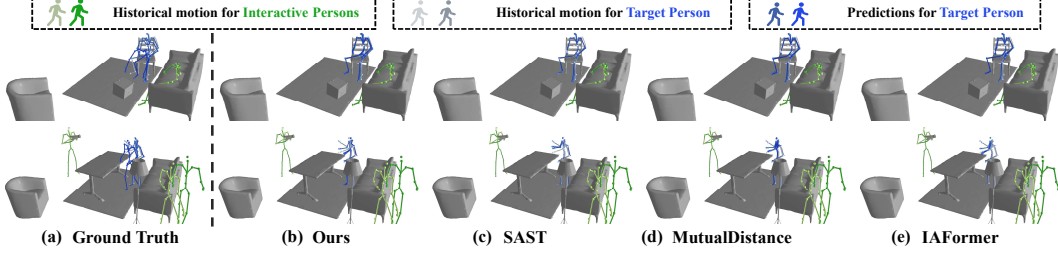

Figure 10: Some failure cases. In the upper example, all methods fail to predict the bending-over action with abrupt motion changes. In the lower example, all methods incorrectly predict that the target person will continue standing by the desk, while the person unexpectedly starts walking away.

**The Monotonous Interaction Modeling Paradigm**  In our work we mainly adopt a monotonous modeling paradigm i.e., the distance-based modeling, where we concatenate distance-based features with intrinsic scene and human features (such as self-encoding) to serve as interaction features.

Euler distance is a natural way to model interaction, as different types of interactions follow distinct distance patterns, even in complex motions like dance. For example, in waltz or tango, the distance between dancers' bodies exhibits repetitive patterns of approaching and retreating. Distance-based modeling captures these patterns, enabling action recognition and future motion prediction. While this modeling paradigm is effective and efficient, exploring alternative paradigms and integrating diverse approaches could lead to more robust and generalized performance.

**Improvement of Joint Multi-person Predictions** We have demonstrated our scalability in joint multi-person forecasting in the paper. However, in our current approach, the model is not explicitly aware of future interactions between individuals, as the input to the HHI module consists only of the historical motion sequence $\mathcal{Y}^{(\mathbf{k})}$. In the future, we could extend the method to an iterative approach where, in each iteration, the HHI module can take the historical motion sequence $\mathcal{Y}^{(\mathbf{k})}$ concatenated with the predicted motion from the previous iteration. This way, when inferring each individual, the model is aware of the predicted future motion of other individuals from the prior iteration via HHI. We leave this extension for future exploration.

## L INVESTIGATION ON SE(3) REPRESENTATIONS

To investigate whether explicit geometric transformations could enhance our model, we implemented the SE(3) relative encoding similar to Miyato et al. (2024) and compared it with our original distance-based encoding. Specifically, we constructed a local coordinate system for each individual in every frame. We defined the root joint (pelvis) as the origin, the vector from the pelvis to the neck as the vertical axis, and the vector between the left and right hips to determine the lateral axis. The forward direction was then derived via the cross product to complete the orthonormal basis. Based on this local frame, we calculated the relative SE(3) transformation between the target and interactive persons. We encoded the rotation using a 6D continuous representation and concatenated it with the original relative XYZ translation coordinates to form the interaction feature. As shown in Table 15, the comparison reveals that the SE(3) encoding achieves performance on par with our original distance-based method. We hypothesize that the lack of improvement suggests that our model is already capable of implicitly learning the necessary geometric relationships and orientations from the temporal patterns of interactive distances. The sequence of distances over time contains rich information about relative motion and heading, which our hierarchical interaction reasoning module effectively captures.

Table 15: Comparison between SE(3) encoding and our distance-based encoding on the HOI-M$^3$ dataset.

| Variant | Mean Path Error (mm) ↓ | Mean Pose Error (mm) ↓ |
|---|---|---|
| Ours | 174.6 | 117.9 |
| Ours + SE(3) | 174.6 | 118.1 |

## M PRELIMINARY VALIDATION ON A SUBSET OF DYNAMIC SCENES

As stated in Line 435, current mainstream datasets lack significant and diverse dynamic scene elements, which limits large-scale benchmarking. However, to validate our architectural claim (that our HSI module can natively handle time-dependent scene coordinates), we conducted an additional experiment on a specific subset of the HOI dataset that contains dynamic objects (e.g., passing or moving an object). Since these samples are rare, we upsampled them during training.

Table 16: Quantitative results on a subset of dynamic objects from the HOI dataset.

| Method | Mean Path Error (mm) ↓ | Mean Pose Error (mm) ↓ |
|---|---|---|
| MutualDistance | 285.0 | 187.9 |
| IAFormer | 288.3 | 176.0 |
| SAST | 281.2 | 189.1 |
| **Ours** | **244.8** | **159.4** |

As shown in Table 16, while the quantitative errors are significantly higher compared to completely static scenes (mainly due to the scarcity of training samples) as expected, our method consistently outperforms baselines in this challenging setting. We demonstrate one successful example at the end of the supplementary video. This confirms our potential to handle dynamic scene elements and suggests that performance would likely improve significantly given a dataset with significant and diverse dynamic scene elements.

## USE OF LARGE LANGUAGE MODELS

During the preparation of this manuscript, large language models (LLMs) were used exclusively as a writing-assistance tool. In particular, LLMs assisted in:

- **Language Polishing**: Improving grammar, sentence structure, and readability while preserving the technical accuracy of the content.
- **Terminology and Style Consistency**: Ensuring consistent usage of technical terms and notation across sections.

LLMs were not involved in the conception of the research problem, the design of the HUMOF framework, or the analysis of results. All scientific contributions are solely the original work of the authors. The LLM was employed only to improve the clarity and presentation of the manuscript's text.

