# OpenReview forum: "HUMOF: Human Motion Forecasting in Interactive Social Scenes"
_ICLR.cc/2026/Conference — ICLR 2026 Poster_

### Official Review · Reviewer_9FNW · 2025-10-29

**Soundness:** 3
**Presentation:** 3
**Contribution:** 3
**Rating:** 6
**Confidence:** 3

**Summary:**

The paper proposes a novel approach to address the problem of human motion forecasting in interactive environments.  It builds a hierarchical interaction representation, modeling interactions at multiple levels by combining explicit cues (e.g. inter-object distances) with learned features. Specifically, HUMOF captures high-level context and low-level details for both social interactions and scene contacts. A key innovation is the coarse-to-fine reasoning module, a multi-layer Transformer that progressively incorporates interaction features: early layers use high-level features, while later layers integrate fine-grained features. An adaptive DCT‐based rescaling further suppresses high-frequency components in early layers, encouraging a coarse-to-fine refinement of the motion prediction.

HUMOF obtained state-of-the-art results on two datasets with human-human and human-scene interactions (HIK. HOI-M3) as well as two datasets with human-scene interactions (GTA-IM, HUMANISE).

**Strengths:**

1. The paper is well written and explains deeply the architecture and how it works.
2. The paper models both human-human and human-scene interactions in a single framework, addressing a realistic scenario of interactive environments.
3. The use of multi-level representations (body-level vs joint-level for social cues, and multi-scale point clouds for scene context) is a strong idea. It balances global context and local detail effectively.
4. The injection of high-level features in early Transformer layers and finer details later (along with the DCT rescaling mechanism) is novel and well motivated by the ablation studies.
5. The authors compare their approach to a large set of baselines and include visualizations to support their claims.

**Weaknesses:**

1. HUMOF’s architecture is quite elaborate (multiple modules, Transformer layers, DCT processing). This complexity might make it hard to reproduce or tune and the paper should give more details in the appendices.
2. The authors note that existing datasets have few moving scene objects. Thus, HUMOF’s performance on truly dynamic environments (e.g. moving furniture or vehicles) is untested.
3. The baselines weren’t supposed to work with all the tested datasets, hence they had to be adapted. This raises a question of fairness in comparisons: it’s possible some methods could be improved with similar context. However, the large gaps suggest HUMOF’s advantage is likely genuine.
4. The runtime analysis in the Appendix was performed on only one dataset (HOI-M3).
5. The supplementary video lacks failure cases. Although some failure scenarios are described in the Appendix, it would be helpful to include them in the video as well.

**Questions:**

1. The paper uses a function $\phi (\cdot)$ to map distances to higher values for closer points. How sensitive is the performance to the choice of this mapping? Could a learned function improve results?
2. HUMOF 3D uses poses for all humans and a 3D point cloud of the scene. In real-world settings, these could be noisy or incomplete. Would the approach still work with noisy or incomplete inputs?
3. The runtime is around 43ms per inference for HOI-M3. Does this runtime change on other datasets? Have the authors tested HUMOF in an online setting? Is the latency competitive for real-time applications?

---

> ### Author Response · Authors · 2025-11-24
> **This is Part 1/2 of our reply**
>
> We sincerely thank the reviewer for the detailed review. We have revised the manuscript accordingly and make the following point-to-point response. Due to word count limitations, we will answer all your questions in two parts. **Below is the first part.**
>
> ## 1. Implementation Details
> We have included further implementation details in Appendix A. We will release our code and weights upon publication to ensure reproducibility as promised.
>
> ## 2. Performance in Environments with Dynamic Objects
> We thank the reviewer for pointing this out. As mentioned before (Line 435 in main paper), current mainstream datasets lack significant and diverse dynamic scene elements, which limits large-scale benchmarking.
> However, to validate our architectural claim (that our HSI module can **natively handle time-dependent scene coordinates**), we conducted **an additional experiment on a specific subset of the HOI dataset that contains dynamic objects** (e.g., passing or moving an object). Due to the scarcity of these samples, we upsampled them during training.
>
>
> | Method                                                                       | Path Err(mm)↓ | Pose Err(mm)↓ |
> | ---------------------------------------------------------------------------- | --------- | --------- |
> | MutualDistance                                                               | 285.0     | 187.9     |
> | IAFormer                                                                     | 288.3     | 176.0     |
> | SAST                                                                         | 281.2     | 189.1     |
> | Ours                                                                         | 244.8     | 159.4     |
>
> As shown in the table above, while the quantitative errors are significantly higher compared to completely static scenes (mainly due to the scarcity of training samples) as expected, our method consistently outperforms baselines in this challenging setting.
> We demonstrate one successful example at the end of the supplementary **Video**.
> This confirms our potential to handle dynamic scene elements and suggests that performance would likely improve significantly given a dataset with enough diverse dynamic scene elements.
>
> We have included these results in Appendix Section M.
>
>
> ## 3. Fairness of Baseline Comparisons
> Adapting baselines to a new task is inherently challenging. We made a best-effort attempt to provide the baselines with the same level of information our model receives. For instance, scene-aware methods were adapted to handle multiple people using our HHI module.
> We agree that while it is possible that bespoke adaptations could further improve baseline performance, the significant performance gap across four different datasets suggests that the advantage of our approach is genuine.
>
> ## 4. Inclusion of Failure Cases
> We have included failure cases from the HIK dataset in the supplementary video (1:41 to 1:56).

---

> ### Author Response · Authors · 2025-11-24
> **This is Part 2/2 of our reply**
>
> ## 5. Distance Mapping Function
> We thank the reviewer for the valuable suggestion.
> We compare the default Gaussian mapping with three variants: Inverse Square, Tanh, and a Learned Residual MLP. The results (Mean Path Error / Pose Error) are shown below:
>
> | Mapping Function | Formula $\phi(d)$                       | Path Error(mm)↓ | Pose Error(mm)↓ |
> | :--------------- | :--------------------------------------- | :------: | :------: |
> | Gaussian (Ours)  | $e^{-d^2 / (2\sigma^2)}$                 |  174.6   |  117.9   |
> | Inverse Square   | $\frac{1}{1 + (d/\sigma)^2}$             |  174.2   |  118.2   |
> | Tanh             | $1 + \tanh(-d/\sigma)$                   |  174.8   |  117.4   |
> | Learned MLP      | $e^{-d^2 / (2\sigma^2)} + \text{MLP}(d)$ |  173.8   |  118.3   |
> |                  |                                          |          |          |
>
> These variants achieve comparable performance, demonstrating that the framework is insensitive to the specific choice of mapping function.
>
> ## 6. Robustness to Noisy and Incomplete Inputs
> In Appendix Section D.4, we have demonstrated our performance under incomplete inputs (missing scene point clouds and interacting individuals).
> To provide a more comprehensive evaluation, we conducted additional experiments using varying levels of noisy inputs (injecting Gaussian noise into the joints of other individuals and the scene point cloud) and incomplete inputs.
> For the latter, we simulated increasing scene occlusion severity by randomly generating 2 to 16 cones originating from the target person's root and removing the point cloud within these cones.
> As illustrated in Figure 6 and Figure 7 in Appendix Sections D.4 and D.5, while performance naturally degrades for all methods as noise or occlusion levels increase, HUMOF consistently outperforms the baselines. This demonstrates that our approach remains effective and robust even under imperfect conditions.
>
> ## 7. Runtime Analysis and Online Applicability
> Thank you for the question. We have evaluated the inference runtime across four datasets. As shown in the table below:
>
> | Method | HOI-M3 | HIK | Humanise | GTA-IM |
> | :--- | :---: | :---: | :---: | :---: |
> | MutualDistance | 196ms | 185ms | 252ms | 158ms |
> | IAFormer | 69ms | 76ms | - | - |
> | SAST | 2s | 2s | 3s | 2s |
> | Ours  | 43ms | 47ms | 97ms | 28ms |
>
> Our method achieves inference times under 100ms per sample in the worst case, corresponding to an inference speed of >10 FPS. This frame rate and latency should be sufficient for most real-time online applications.

---

### Official Review · Reviewer_sVzv · 2025-10-31

**Soundness:** 3
**Presentation:** 3
**Contribution:** 2
**Rating:** 6
**Confidence:** 5

**Summary:**

The paper introduces HUMOF, a novel framework for human motion forecasting in complex social environments. HUMOF effectively integrates human kinematics, dynamics, spatial–temporal context, and interaction cues into a unified predictive model.

A key contribution is the Hierarchical Interaction Representation, which jointly captures human–human and human–scene interactions using both explicit distance features and learned semantic–geometric representations. To exploit this representation, the authors design a Coarse-to-Fine Interaction Reasoning Module, which improves motion prediction through two mechanisms:

Spatial hierarchy: High-level semantic features are introduced in early Transformer layers, while fine-grained geometric cues are progressively refined in later layers.

Frequency control: A DCT-based frequency modulation strategy prioritizes high-frequency motion components early in training and focuses on low-frequency refinements in later stages.

**Strengths:**

1. The paper is logically organized and easy to follow, with a coherent narrative from motivation to methodology and results.

2. The evaluation covers both multi-person interactive and single-person forecasting scenarios, including tests on unseen environments. The proposed method consistently outperforms baselines across all benchmarks.

3. The visualizations and supplementary videos are convincing — showing smooth, stable, and realistic motions with minimal jitter and drift compared to prior works.

**Weaknesses:**

No major weakness

**Questions:**

N/A

---

> ### Author Response · Authors · 2025-11-24
>
> We sincerely thank the reviewer for the positive assessment and for recognizing the effectiveness of our Hierarchical Interaction Representation and Coarse-to-Fine Interaction Reasoning Module. We are encouraged that the reviewer found our paper logically organized, the evaluation comprehensive, and the visualizations convincing.

---

### Official Review · Reviewer_ztD4 · 2025-10-31

**Soundness:** 2
**Presentation:** 3
**Contribution:** 3
**Rating:** 6
**Confidence:** 4

**Summary:**

HUMOF presents a human motion forecasting method in social environments that takes into account both human-human as well as human-scene interactions. A DCT rescaling allows for controllability of the coarseness of the signal processing. The authors compare on two social scene datasets and on two human-scene interaction datasets and consistently outperform SOTA.

**Strengths:**

The problem of social human motion forecasting is highly relevant but relevantly under-explored. HUMOF is a complex method with many moving parts. However, the authors ablate the model relatively well. Utilizing DCT is a commonly used technique in motion forecasting. Using the rescaling for coarse-to-fine prediction in the context of human motion fc is novel and clever. The method description is mostly clear experiments have been conducted on two social scene datasets and two human-scene interaction datasets, demonstrating the methods effectiveness, while being parameter and inference time efficient.

**Weaknesses:**

The method is complex (as the task itself is complex) but I have some concerns about some of the model specifics:
(1) The utilized relative encoding is overly simplistic: I wonder if instead of just utilizing the point distance the method could utilize the geometric transformations in SE(3), for example as has been utilized in [1].
(2) While DCT works well for-single person motion, I wonder if it is limiting the human-object and human-human interaction quality. In the transformer, the tokens do not directly correspond to frames anymore, so temporal alignment over longer time frames might be hindered. The authors should evaluate this by showing either closer human-human (or human-object) interaction.
If the current datasets do not contain sufficient close person-to-person data, the authors should utilize other dyadic datasets, i.e Inter-X [2].

My second concern is with regards to the evaluation: the authors only evaluate directly comparing to GT (path error, pose error) - however, due to the complexity of the scene, multiple “answers” could be correct - I wonder if the authors have considered utilizing methods to compare the generated sequences on a distribution level, i.e. by utilizing FID, i.e. the authors could use the combined input-output sequence ($X^{1:H} \oplus \hat{X}^{H+1:H+T}$) to compare the distributions.

[1] GTA: Geometric Transform Attention (ICLR 2024)

[2] Inter-X: Towards Versatile Human-Human Interaction Analysis (CVPR 2024)

**Questions:**

Did the authors measure human-object and human-human penetration? This feels like a natural form of evaluating this task.

---

> ### Author Response · Authors · 2025-11-24
> **This is Part 1/2 of our reply**
>
> We are very thankful to your valuable suggestions and the careful reading. We have revised the manuscript accordingly and make the following point-to-point response. Due to word count limitations, we will answer all your questions in two parts. **Below is the first part.**
>
> ## 1. SE(3) Geometric Representations
> Thanks for this constructive suggestion. To investigate whether explicit geometric transformations could enhance our model, we implemented SE(3) relative encoding and compared it with our original distance-based encoding.
>
> Implementation Details: we constructed a local coordinate system for each individual in every frame. We defined the root joint (pelvis) as the origin, the vector from the pelvis to the neck as the vertical axis, and the vector between the left and right hips to determine the lateral axis. The forward direction was then derived via the cross product to complete the orthonormal basis. Based on this local frame, we calculated the relative SE(3) transformation between the target and interactive persons. We encoded the rotation using a 6D continuous representation and concatenated it with the original relative XYZ translation coordinates to form the interaction feature.
>
> As shown below, the comparison reveals that the SE(3) encoding achieves performance on par with our original distance-based method.
>
> | Variant | Path Error (mm) $\downarrow$ | Pose Error (mm) $\downarrow$ |
> | :--- | :---: | :---: |
> | Ours | 174.6 | 117.9 |
> | Ours + SE(3) Encoding | 174.6 | 118.1 |
>
>  We hypothesize that the lack of significant improvement suggests that our model is already capable of implicitly learning the necessary geometric relationships and orientations from the temporal patterns of interactive distances. The sequence of distances over time contains rich information about relative motion and heading, which our hierarchical interaction reasoning module effectively captures.
> We found this investigation highly valuable and have added a detailed discussion and these results to Appendix Section L.
>
> ## 2. Impact of DCT on Closer Interaction
> We understand the concern regarding the impact of DCT on close interaction quality. While DCT tokens do not correspond to individual frames, they encode the global temporal evolution of the motion sequence. This global view preserves the complete motion context, sufficient for modeling coherent interactions.
>
> To empirically verify whether the DCT representation limits close interaction quality, we evaluated our model on a subset of the HOI dataset focusing specifically on **close interactions**, where the target person is in close proximity ($<15$cm) to scene objects or other individuals.
>
> | Test Set | Path Error $\downarrow$ | Pose Error $\downarrow$ |
> | :--- | :---: | :---: |
> | Ours (Full Test Set) | 174.6 | 117.9 |
> | **Ours (Close Interaction Subset)** | **172.5** | **116.8** |
>
> As shown above, the DCT-based method does not limit interaction quality. The results on this subset are even slightly better than the full test set. This is likely because proximal scene objects or other individuals provide stronger geometric constraints on the target motion. Thus, it becomes easier for the model to make better predictions compared to samples where there are no very close scene objects or other persons.
>
> We have added these results to a new section in the Appendix (Section G). We also provide visual examples of these close human-human (InterX dataset [2]) and human-scene (HUMANISE dataset) interactions in the Supplementary **Video (0:11 to 0:41)** to further demonstrate the high interaction quality.
>
> [2] Inter-X: Towards Versatile Human-Human Interaction Analysis (CVPR 2024)

---

> ### Author Response · Authors · 2025-11-24
> **This is Part 2/2 of our reply**
>
> ## 3. Distributional Evaluation (FID)
> We thank the reviewer for this suggestion. Following your advice, we utilized the pre-trained motion encoder from T2M [3] as a feature extractor to map motion sequences into the feature space (512-dim) and then calculated the FID on the HOI dataset.
>
> The results are summarized below:
>
> | Method   | FID ($\downarrow$) |
> | :------- | :----------------: |
> | SAST     |       0.0278       |
> | MutualDistance     |       0.0233       |
> | IAFormer |       0.0170       |
> | **Ours** |     **0.0164**     |
>
> Our method achieves the lowest FID score, indicating that our generated motions better match the ground-truth distribution than the baselines.
>
>
>
>
>
> ## 4. Penetration Metrics
> We calculated the mean penetration rate and penetration depth on the HOI-M3 dataset, which provides the high-quality scene meshes and human body models (SMPL-X) necessary for this analysis. The metrics are defined as follows:
>
> - Penetration depth at frame $t$ (in meters) is defined as the sum of absolute signed distance field (SDF) values for all joints of the target person that penetrate the scene or other persons:
> $\sum _ {j=1}^{J}\left|\left(\Psi\left(\mathbf{X} _ {j}^{t}\right)\right) _ {-}\right|$, where $\Psi(\cdot)$ denotes the signed distance field (SDF) of the scene or interactive persons, $(\cdot) _ {-}$ clips all positive distances to zero, and $\mathbf{X}^t _ j$ is the 3D position of the $j$-th joint at time $t$.
>
> - Penetration rate at frame $t$ is the ratio of joints with a negative SDF value to the total number of joints:
> $\frac{\text { Number of joints with } \Psi\left(\mathbf{X} _ {j}^{t}\right)<0}{J}$
>
> We then take the average over the frame and sample dimensions to obtain the final mean metrics.
>
> **Penetration Results on the HOI-M3 dataset**
>
> |                        | **Human-to-Scene**    |                               | **Human-to-Human**    |                               |
> | ---------------------- | --------------------- | ----------------------------- | --------------------- | ----------------------------- |
> |                        | Mean Penetration Rate | Mean Penetration Depth (mm) | Mean Penetration Rate | Mean Penetration Depth (mm) |
> | Ground-Truth (GT)      | 1.84%                 | 11.26                         | 0.070%                | 0.61                          |
> | SAST                   | 1.54%                 | 11.07                         | 0.063%                | 0.45                          |
> | IAFormer               | 1.62%                 | 10.89                         | 0.068%                | 0.59                          |
> | MutualDistance         | **1.45%**                 | **10.68**                         | 0.057%                | 0.41                          |
> | Ours                   | 1.49%                 | 10.77                         | **0.052%**                | **0.39**                          |
>
> As shown above, MutualDistance achieves the best HSI penetration scores, likely due to its use of mesh-based modeling for both the target person and the scene, providing more explicit surface information to avoid penetration. However, we note that penetration metrics alone, without considering motion accuracy, should be interpreted with caution. Our case analysis reveals that higher penetration rates can sometimes result from accurately predicting dynamic motion, whereas lower penetration may occur when a method predicts static or incorrect motion (e.g., the person remaining stationary). This is also supported by the fact that the Ground-Truth (GT) motion itself registers the highest penetration, due to small misalignments between the motion capture data and scanned scene geometry in the dataset, whereas the four prediction methods may inadvertently avoid penetration by under-predicting movement (e.g., remaining static). Our method strikes a strong balance, achieving high motion accuracy while maintaining penetration scores comparable to or better than most baselines.
>
> To further reduce penetration while preserving accuracy, we could also adopt mesh-based modeling—e.g., after computing joint-to-point/joint distance, we then adjust these by subtracting the point-to-mesh surface distance to get mesh-to-mesh distances. In our current method we did not adopt this design as this would introduce dependencies on scene meshes and SMPL-X parameters, increasing complexity and reducing practical applicability.
>
>
> [2] Inter-X: Towards Versatile Human-Human Interaction Analysis (CVPR 2024)
>
> [3] Generating Diverse and Natural 3D Human Motions from Text (CVPR 2022)

---

### Official Review · Reviewer_7peC · 2025-11-02

**Soundness:** 3
**Presentation:** 2
**Contribution:** 3
**Rating:** 6
**Confidence:** 4

**Summary:**

This paper proposes HUMOF, a method for human motion forecasting in complex dynamic scenes that involve both human–human and human–scene interactions.
The approach introduces a hierarchical interaction representation that separately encodes high-level contextual and low-level geometric information, and a coarse-to-fine interaction reasoning module that injects these features into Transformer layers from semantic to geometric levels.
The framework combines Discrete Cosine Transform, Graph Convolutional Networks, and Transformer architectures.
Experiments on four public datasets (HIK, HOI-M3, GTA-IM, and HUMANISE) show that HUMOF achieves state-of-the-art performance.

**Strengths:**

1.	Comprehensive interaction modeling.

- Unlike prior works that focus on either human–scene or human–human interactions, this paper successfully integrates both within a unified framework. This joint modeling is meaningful and leads to noticeable performance improvements on dynamic scenes.

2.	Strong empirical performance.

- The method achieves consistent improvements across multiple datasets and evaluation metrics. The results demonstrate that the hierarchical representation and coarse-to-fine reasoning work effectively together.

3.	Practical contribution as a baseline.

- If the authors release the code as promised in the abstract, HUMOF can become a valuable benchmark for future research on interactive human motion prediction.

**Weaknesses:**

1.	Limited novelty in components.

Most elements of the architecture come from existing works (coarse-to-fine approach, distance-based interaction modeling, abstraction, ...). The contribution lies primarily in how these components are integrated, rather than in introducing a fundamentally new modeling paradigm.

2.	Lack of quantitative evaluation for multi-person inference.

The paper briefly demonstrates qualitative results for joint multi-person inference in Figure 4 but does not provide quantitative metrics or runtime analysis. Since multi-person prediction is highly relevant for real-world applications such as social robotics or crowd simulation, the absence of measurable evaluation limits the practical significance of this claim.

3.	High computational complexity for multi-agent prediction.

The proposed framework models each target person independently and computes pairwise interactions with every surrounding person.
This design implies that when there are K individuals, the method requires roughly K² human–human interaction computations.
Furthermore, because the scene abstraction (HSI) is recomputed for each target, the total cost scales linearly with K, resulting in O(K³) level complexity when both factors are considered.
This raises concerns about the scalability and feasibility of real-time multi-agent forecasting.

4.	Issue in references.

The paper repeatedly cites prior works using informal author patterns such as “Jeong & etc., 2024” or “Xing & etc., 2025.” This format is inappropriate for a scientific paper and must be corrected to proper citation styles.

**Questions:**

Please see the weakness section

---

> ### Author Response · Authors · 2025-11-24
>
> We sincerely thank the reviewer for the careful reading and constructive suggestions. We have revised the manuscript accordingly and make the following point-to-point response.
>
> ## 1. Clarification on Novelty of the Framework Design
> While we agree that individual atomic components (set abstraction, distance modeling, etc.) are established techniques, we do not claim novelty in these components individually. Instead, our core novelty lies in **the unified framework specifically designed to address the unique challenges of motion prediction in dynamic scenes**.
>
> The primary challenge in dynamic scene forecasting is effectively capturing environmental elements that are heterogeneous in form: scene objects (point clouds) and humans (joints sequences). A naive solution encodes these separately and fuses them "flatly" (SAST). However, this approach is limited as it (1) encodes environmental elements themselves rather than explicit interactions, and (2) ignores the natural hierarchical nature of interactions.
>
> To address this, we propose **a unified Hierarchical Framework that bridges both domains**. We enforce consistent design principles by encoding both Human-Human and Human-Scene interactions using a Hierarchical Interaction Representation based on interactive distances. Coupling this with our Coarse-to-Fine Interaction Reasoning transforms the paradigm from naive, flat fusion to explicit, hierarchical interaction modeling. This introduces a structured inductive bias, ensuring environmental hints and constraints are processed in a logical order—from global semantics to local geometric details—achieving superior performance in dynamic environments.
>
> ## 2. Quantitative Evaluation for Multi-Person Inference
> We thank the reviewer for this question.
> For accuracy evaluation, the metrics used in the multi-person inference setting are identical to those used in the person-by-person inference setting (i.e. Table 1 in main paper).
> For efficiency evaluation, we report the mean inference time on the HOI dataset:
>
> | Method             | MutualDistance | IAFormer | SAST |   Ours   |
> | :----------------- | :------------: | :------: | :--: | :------: |
> | **Inference Time** |     126ms      |   69ms   |  2s  | **45ms** |
>
>
> ## 3. Computational complexity for multi-agent prediction.
> We thank the reviewer for the valuable question. We address this concern from three perspectives: theoretical complexity, practical scalability, and optimization strategies.
>
> **1. Theoretical Complexity ($O(K^2)$ vs. $O(K^3)$):**
> We clarify that the theoretical complexity of our interaction modeling is $O(K^2)$, not $O(K^3)$. In a single batch containing $K$ agents, each agent computes cross-attention with the other $K-1$ agents. Thus, the total number of pairwise interactions computed is proportional to $K^2$.
>
> **2. Practical Scalability:**
> While $O(K^2)$ can be computationally intensive, the practical complexity is manageable for the following reasons:
> *   **Locality of Interaction:** For short-term prediction, considering interactions within a 3.5m radius is sufficient. In real-world applications, it is extremely rare for more than 100 people to occupy such a space.
> *   **Stress Test Results:** We evaluated the average inference time for varying crowd densities (from 20 to 100 persons) on a Titan RTX (24GB). The results demonstrate that the runtime scales acceptably even at high densities:
>     *   20 persons: 0.09s
>     *   40 persons: 0.17s
>     *   60 persons: 0.26s
>     *   80 persons: 0.36s
>     *   100 persons: 0.52s
>
> **3. Further Mitigation via Token Drop:**
> To further reduce latency in crowded scenes, we exploit the sparsity of human interactions (a target individual usually interacts with only a limited number of neighbors).
> *   **Method:** Most computational cost arises from the late transformer layers where the context consists of fine-level tokens (where each person contributes $J$ tokens). Leveraging our **coarse-to-fine reasoning strategy**, we utilize intermediate states in the early layers where each context token corresponds to one individual. We feed the attention map and the token of the other individual into a lightweight MLP-based Gate to generate an interaction score. In the subsequent late layers, we drop the fine-level tokens corresponding to individuals with low scores, who are unlikely to interact with the target.
> *   **Results:** Applying this acceleration strategy on the HOI dataset reduced inference time from 45ms to 37ms with negligible performance drop, as shown below. We believe this acceleration will be even more significant in denser scenes.
>
> | Method            |   Time   | Path Error | Pose Error |
> | :---------------- | :------: | :--------: | :--------: |
> | Ours              |   45ms   |   174.6    |   117.9    |
> | Ours + Token Drop | **37ms** |   174.6    |   118.1    |
>
> ## Incorrect reference format
> We appreciate for pointing this out and have corrected the citation errors in the updated paper.

---

### Author Response · Authors · 2025-12-01
**Rebuttal Summary for the Area Chair**

Dear Area Chair,

We thank the reviewers for their positive feedback and valuable suggestions. Below is a summary of how we addressed the key points:

**Efficiency**
*   Complexity of Multi-agent Batch Inference: Stress tests with crowd sizes up to 100 agents demonstrate that the practical complexity is acceptable.
*   Runtime on four datasets: In the worst-case scenario our method still achieves >10 FPS.

**More Comprehensive Evaluation**
*   Added FID scores and penetration metrics.
*   Close interaction: Our method maintains quality on close interaction subset (<50cm).
*   Robustness: The model remains robust under imperfect (noisy or incomplete) inputs, outperforming baselines.

**Extension**
*   Additional experiments on dynamic scene objects subsets, outperforming baselines and validating the potential for dynamic scene objects.

---

### Meta-Review · Area_Chair_RHRW · 2026-01-05

**Summary:**

The reviewers appreciate the unified framework proposed in this paper and its empirical evaluation, but nonetheless express some concerns regarding the complexity of this framework, the novelty of its component, and some aspects of the evaluation, including computational complexity in the multi-agent case, FID and penetration as metrics, impact of DCT on close interactions, robustness to noise, runtime analysis.

**Reviewer Concerns:**

All the concerns related to empirical evaluation were convincingly addressed by the authors in their responses. The comments related to the complexity of the framework and the novelty of its components were also addressed, although these questions are often quite subjective.

**Reviewer Scores:**

Altogether, the reviewers' scores were already on the positive side and would have likely further increased if the reviewers had been able to participate fully in the discussion.

---

### Decision · Program_Chairs · 2026-01-26

Accept (Poster)